# AUTOMATED DISCOVERY OF PAIRWISE INTERACTIONS FROM UNSTRUCTURED DATA

## ABSTRACT

Pairwise interactions between perturbations to a system can provide evidence for the causal dependencies of the underlying underlying mechanisms of a system. When observations are low dimensional, hand crafted measurements, detecting interactions amounts to simple statistical tests, but it is not obvious how to detect interactions between perturbations affecting latent variables. We derive two interaction tests that are based on pairwise interventions, and show how these tests can be integrated into an active learning pipeline to efficiently discover pairwise interactions between perturbations. We illustrate the value of these tests in the context of biology, where pairwise perturbation experiments are frequently used to reveal interactions that are not observable from any single perturbation. Our tests can be run on unstructured data, such as the pixels in an image, which enables a more general notion of interaction than typical cell viability experiments, and can be run on cheaper experimental assays. We validate on several synthetic and real biological experiments that our tests are able to identify interacting pairs effectively. We evaluate our approach on a real biological experiment where we knocked out 50 pairs of genes and measured the effect with microscopy images. We show that we are able to recover significantly more known biological interactions than random search and standard active learning baselines.

## 1 INTRODUCTION

Across the sciences, measurement of pairwise interaction between perturbations often reveals the existence of underlying mechanisms that single perturbations cannot. For example, quantum entanglement experiments reject classical laws by showing that entangled particles' spins are perfectly anti-correlated once we make a measurement, when classical laws predict independence. In economics, people may be more willing to pay for goods when presented in a bundle than they are to pay for each good in isolation, which reveals complements in the underlying consumer preferences. And in biology—the application area on which we focus—the concept of *synthetic lethality* (Nijman, 2011) occurs when the simultaneous perturbation of two genes results in cell death while individual perturbations do not, thereby revealing complementary roles in the underlying cellular mechanisms.

To demonstrate a pairwise interaction a scientist has to carefully design three steps of the experiment:

1. **Measurement:** the expert to selects properties of the system to measure to reveal an interaction. E.g., measuring cell viability may reveal synthetic lethality, while the cell's colour will not.
2. **Hypothesis testing:** an interaction is a deviation from what we expect under a null hypothesis that assumes independent effects. The expert needs to predict the outcome under independence and compare this prediction to the actual outcome.
3. **Selection:** there are typically many variables that one could perturb, but only a small fraction will exhibit surprising interactions. The expert selects pairs of perturbations from all possible pairs.

Measuring interactions is further complicated by the fact that measurement and hypothesis testing are interdependent. We can see this in synthetic lethality example, where if we choose to measure the fraction of cells that survived the respective perturbations, then in order to test independence we would need to test whether $P(\text{survive}_a \cap \text{survive}_b) = P(\text{survive}_a)P(\text{survive}_b)$. If we had instead measured the fraction of cells that died, we would have needed to test whether $P(\text{die}_a \cup \text{die}_b) = P(\text{die}_a) + P(\text{die}_b) - P(\text{die}_a)P(\text{die}_b)$. The state of the cells in the respective petri dishes is the same in both cases, but correctly testing (in)dependence depends on how the expert chose to measure

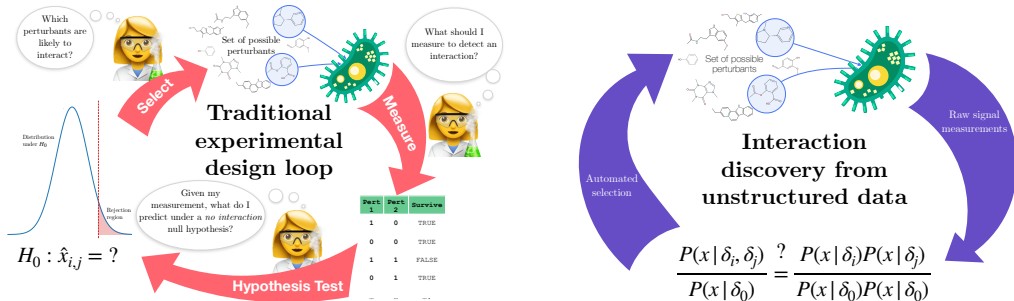

Figure 1: The traditional experimental design relies on human experts to choose features of the experimental outcome (e.g., did a cell survive a perturbation?), predict behavior under the null hypothesis to test for interactions, and select interacting pairs from a combinatorial space of perturbations. Our approach automates the selection of perturbants and interaction testing directly from raw signal data (e.g., microscope images of cells).

that state. These three steps require significant expertise and knowledge to choose and conduct the measurements, which makes scientific discovery hard to automate and scale.

Modern high throughput screening platforms (Dove, 2007; Baillargeon et al., 2019; Blay et al., 2020; Bock et al., 2022; Fabio et al., 2023; Morris et al., 2023; Messner et al., 2023) enables running large scale perturbation experiments that collect information-rich unstructured measurements. For example, cell painting assays (Bra, 2016; Fay et al., 2023; Chandrasekaran et al., 2023) provide microscopy images of cells, capturing the same information a human expert would measure, but without pre-selecting a specific property like cell viability. This lets us *measure* all the potential properties of interest at once—albeit as unstructured signal data, such as raw pixel images or sensor measurements, rather than preprocessed properties—but it is not clear how to use these measurements for interaction *testing* and *selection* in order to efficiently discover pairwise interactions. This suggests the primary question that this paper seeks to address: if we dispense with prespecified structured measurements, can we automate the discovery of pairwise interactions by designing testing and selection procedures that efficiently detect interactions from unstructured data?

To test for interactions, we show that pairwise perturbations are *separable* if they provide no additional information beyond what is already known from the single perturbations. This separability property can be tested by comparing density ratios of the single and double perturbations to the control distribution. We also develop a test for the case where pairs of perturbations affect disjoint subsets of the outcome space. For example, if two perturbations affect different organelles in the cell, each of which affect disjoint pixels in an image. This strong form of non-interaction allows us to compose summary statistics from single perturbations to predict the corresponding statistics from double perturbations. In particular, this lets us compose embeddings of single perturbations to predict the embeddings of double perturbations, similar to word-vector analogies Mikolov et al. (2013).

With this notion of an interaction, we address the second task of selection via efficient experimental design. We can search the space of pairwise experiments by selecting pairs of perturbations that are likely to result in large test statistics. In doing so, we reduce the problem of finding interacting pairs of perturbations into an active matrix completion problem. By defining this notion of interaction, we avoid the need to characterize uncertainty over the pixel-level outcomes (or some embedding thereof). Instead we directly model the posterior over an unknown reward matrix defined by the test statistics and at every round select actions striking a balance between exploration and exploitation via information directed sampling (IDS; Russo & Van Roy, 2016; Xu et al., 2022).

We evaluated our approach on both synthetic and real biological experiments. On the synthetic tasks, we found that our tests were able to detect both forms of dependence, validating our theory. On a benchmark consisting of all pairs of gene knockouts for 50 genes in HUVEC cells, we found that our approach using IDS discovers pairs of genes with higher interaction scores significantly faster than baselines, resulting in $15\%$ more known biological interactions being discovered and potentially far more novel interactions. The interactions we detected were also complementary to those which would have been discovered using existing cosine similarity-based approaches, and as a result, the two approaches can be combined to get a more detailed estimate of the relationships between genes from

perturbation experiments. In summary, we formulate two notions of pairwise interaction—violations of separability and disjointedness—and develop a system for discovering interactions through the following contributions:

- We show that two perturbations are separable if double perturbations provide no additional information than that which was revealed in single perturbations.
- Further, we show that two perturbations are disjoint if and only if their densities compose additively. This result also implies sufficient conditions for embeddings of perturbations to be composed additively to predict pairwise perturbations.
- We demonstrate that the test scores for detecting interactions can be used to efficiently search for pairs of perturbations that interact using active matrix completion algorithms.

## 2 RELATED WORK

**Causal representation learning** Our approach to the problem of detecting interactions builds on the modelling assumptions developed in nonlinear independent component analysis (Hyvärinen & Pajunen, 1999; Hyvarinen & Morioka, 2016; Hyvarinen et al., 2019) and causal representation learning (Schölkopf et al., 2021), where we assume we observe some nonlinear mixing function, $g(\cdot)$, of the underlying latent variables. The causal representation learning literature typically focuses on disentangling latent variables, whereas we look for testible implications without disentanglement. Our data generating process assumptions are most similar to the interventional setting (Ahuja et al., 2023; Squires et al., 2023; Buchholz et al., 2023; Varıcı et al., 2024). If we could successfully disentangle latent variables, then our task would be straightforward, but disentanglement is challenging in practice because it is not possible to validate whether an algorithm has succeeded in disentangling latent variables without access to ground truth. Like us, Jiang & Aragam (2023) also attempt to learn latent dependence properties (they characterize the whole latent DAG, not just marginal dependence) without disentanglement, but they assume a fixed bipartite graph of dependence between latents and observations, which does not apply to the pixel-level observations that we study. Zhang et al. (2023) develop causal representation learning techniques to disentangle latent variables and characterize conditions for extrapolation. Like us, they focus on biological applications, but they rely on stronger polynomial assumptions to achieve disentanglement. Their conditions for extrapolation are complementary to our separability tests: they argue extrapolation is possible when interventions affect non-overlapping latents, while we aim to discover when interventions affect *overlapping* latents. Finally, our separability test is closely related to, and inspired by Wang et al. (2023), but whereas they assume separability of concepts to manipulate generative models, we aim to test for an analogous notion of independence in experimental data.

**Representation learning for gene knockouts** Our experiments build on a number of recent works showing the effectiveness of embedding cells presented into a representation space Sypetkowski et al. (2023); Kraus et al. (2023); Xun et al. (2023). These works show that you can infer that genes code for proteins that form part of complexes by finding embeddings that are highly cosine similar. This works because knocking out proteins on the same pathway will induce the same morphological effect, but this approach is limited to effects that are revealed by single perturbations. There have also been a number of papers that have attempted to learn disentangled representations of cells (e.g. Lotfollahi et al., 2023; Bereket & Karaletsos, 2023; Lopez et al., 2023), mostly from gene expression data.

**Design of Gene Knockout Experiments** There is growing literature on developing algorithms for the design of gene knockout experiments. Typically the problem is studied in the context of discovering single gene-knockouts (Mehrjou et al., 2021) which result in a particular phenotypic effect of interest. A variety of methods including bandits (Pacchiano et al., 2023) and traditional experimental design (Lyle et al., 2023) approaches have been studied in this context. Further, approaches which aim to learn predictors for the effects of unseen gene knockouts typically operate on RNA-seq data (Huang et al., 2023). RNA-Seq data is far more structured than the image data we consider: the former is a high-dimensional vector of count data where each element corresponds to a particular gene's expression, while the latter is just an unstructured collection of pixel intensities.

**Experimental design.** The task of efficiently designing experiments to gain the most information about a system can be formalized as sequential Bayesian optimal experimental design (BOED; Ryan et al., 2016; Foster, 2021; Rainforth et al., 2023), where the goal to design experiments $x \in \mathcal{X}$ with outcomes $y \in \mathcal{Y}$ governed by a generative process $y \sim p(y \mid \gamma, x)$ with parameters $\gamma$. The experiments $(x_1, \ldots, x_T)$ are performed sequentially, with the objective of maximizing the information gain (Lindley, 1956; Sebastiani & Wynn, 2000). However, applying this framework in

high-dimensional settings is challenging because estimating information gain becomes difficult. We sidestep this complexity by focusing on the specific task of detecting interactions.

**Statistical and biological interactions.** We further provide a discussion about prior works about classical statistical interactions in regression models and biological interactions in Appendix A.

## 3 TESTS FOR PAIRWISE INTERACTIONS

In this section we develop tests for both *separable* and *disjoint* interventions. The separability test, Section 3.1, allows us to test whether two perturbations act on disjoint sets of latent variables. Disjointedness, Section 3.2, implies compositional generalization of summary statistics, allowing us to reduce the search space by predicting the outcome of experiments without explicitly running them.

**Setup** We have observations $X$ in an *observation space* $\mathcal{X}$ of unstructured measurements such as pixels in an image, and a finite set of perturbations $\{T_i \in \mathcal{T}_i : i \in [n]\}$. Although our theory accommodates generic perturbations, we restrict the discussion to binary perturbations for convenience, i.e., $\mathcal{T}_i := \{0, 1\}$ for all $i \in [n]$, and $T_i = 1$ means perturbation $i$ is applied. For all $i, j \in [n]$, denote the perturbation indicator as follows:

$$\delta_0 = \{T_{[n]} = 0\} \quad \delta_i = \{T_i = 1, T_{[n]\setminus\{i\}} = 0\} \quad \delta_{ij} = \{T_i = T_j = 1, T_{[n]\setminus\{i,j\}} = 0\}.$$

In this section, we assume that for a pair of perturbations $T_i, T_j$, we have access to experimental data from four distributions: $p(x|\delta_0), p(x|\delta_i), p(x|\delta_j)$, and $p(x|\delta_{ij})$; when we discuss the active learning procedures in Section 4, we will assume that we have access to all single perturbations distributions, $p(x|\delta_i)$, and we will adaptively select the pairs, $i, j$, on which to collect samples from $p(x|\delta_i, \delta_j)$. Finally, we assume the existence of a set of latent variables $\{Z_1, \ldots, Z_L\} \subseteq \mathcal{Z}$ in some latent space, $\mathcal{Z}$, that capture all relevant information about the perturbation. We state this precisely as,

**Assumption 3.1.** $X \perp\!\!\!\perp (T_1, \ldots, T_n)|Z$, *or equivalently,*

$$p(x|T_1, \ldots, T_n) = \int_{\mathcal{Z}} p(x|z)p(z|T_1, \ldots, T_n)\mathrm{d}z.$$

Throughout, we assume all distributions have well-defined densities or probability mass functions with respect to some $\sigma$-finite base measure on their corresponding sample spaces. We use the same symbol to denote a distribution and its density. All proofs are deferred to Appendix B.

### 3.1 SEPARABILITY TESTING

It would be trivial to verify whether two perturbations intervene on different latents if the underlying latent variables were observed directly. However, our observations are unstructured signal data, which we assume is generated by some deterministic mixing function.

**Assumption 3.2.** *There exists a diffeomorphism[1] $g : \mathcal{Z} \to \mathcal{X}$ such that $X = g(Z)$.*

**Remark 3.3.** *This diffeomorphism assumption between $\mathcal{Z}$ and $\mathcal{X}$ can be relaxed; our theory and methodology remain valid as long as the change of variable formula holds for the distributions of $Z$ and $X$. For example, the distribution of $X$ may have support on a low-dimensional manifold of $\mathcal{X}$, with the latent space $\mathcal{Z}$ having a lower dimension than $\mathcal{X}$. See Krantz & Parks (2008, Lemma 5.1.4) for the generalized change of variable formula for non-bijective transformations.*

The key observation that we leverage is that when latent variables are independent, the change of variable formula implies that the density ratio of a perturbed distribution to the original (control) distribution has a simple form that only involves the distribution of the intervened *latent* variable,

$$\frac{p(x|\delta_i)}{p(x|\delta_0)} = \frac{p_Z(g^{-1}(x)|\delta_i)\left|\det(J(g^{-1}(x)))\right|}{p_Z(g^{-1}(x)|\delta_0)\left|\det(J(g^{-1}(x)))\right|} = \frac{p_{Z_i}^{\dagger}([g^{-1}(x)]_i)}{p_{Z_i}([g^{-1}(x)]_i)}.$$

Here $p_{Z_i}^{\dagger}$ denotes the perturbed distribution of the latent variable $Z_i$ targeted by intervention $i$, $[\cdot]_i$ denotes the projection operator to the subspace of $Z_i$. The testable conclusion that we can derive from this observation, is that when two variables are independent, we can predict the density ratio of the double perturbation, $\frac{p(x|\delta_{ij})}{p(x|\delta_0)}$, as the product of the density ratios of the respective single perturbations, $\frac{p(x|\delta_i)}{p(x|\delta_0)}\frac{p(x|\delta_j)}{p(x|\delta_0)}$. To make this rigorous, we assume that there exists a causal factorization of the latent distribution, where each latent variable is conditionally independent of its non-descendents given its parents. We use $\mathrm{Pa}(Y)$ (resp. $\mathrm{Ch}(Y)$) to denote the parent (resp. children) of a random variable $Y$.

---

[1] A diffeomorphism is a differentiable bijection with a differentiable inverse.

**Assumption 3.4.** *The latent distribution $p_Z(z)$ can be factorized into $L$ latent factors:*

$$p_Z(z|T_1, \ldots, T_n) = \prod_{l=1}^{L} p_{Z_l}(z_l | \mathrm{Pa}(z_l)),$$

*where $\forall l \in [L]$, $\mathrm{Pa}(Z_l) \subseteq T_{[n]} \cup Z_{[L]}$, and $\forall i \in [n]$, $\mathrm{Ch}(T_i) \subseteq Z_{[L]}$. The random variable $Z$ here should be interpreted as the concatenation of all latent factors $\{Z_1, \ldots, Z_L\}$.*

The observed effects of two non-interacting perturbations are attributed to distinct causal pathways, rather than being confounded by a shared latent factor.

**Definition 3.5.** *Two perturbations $\delta_i, \delta_j$ are **separable** if $\mathrm{Ch}(T_i) \cap \mathrm{Ch}(T_j) = \emptyset$.*

When two perturbations act separably, the resulting density ratios of the observation distribution will have the predictable interactions that we derived above.

**Theorem 3.6.** *Suppose that Assumptions 3.4 and 3.2 hold, and that $\delta_i, \delta_j$ are separable. Then,*

$$\frac{p(x|\delta_{i,j})}{p(x|\delta_0)} = \frac{p(x|\delta_i)}{p(x|\delta_0)} \frac{p(x|\delta_j)}{p(x|\delta_0)}. \tag{1}$$

This provides a testable implication of separability, but to test it, we need to derive a real-valued test statistics. By taking logs and rearranging terms, we can rewrite Eq. (1), as testing,

$$\log p(x|\delta_{ij}) + \log p(x|\delta_0) - \log p(x|\delta_i) - \log p(x|\delta_j) \overset{?}{=} 0 \tag{2}$$

If we instead take expectations of Eq. (2) with respect to $p(x|\delta_0)$, then this amounts to testing if

$$\mathrm{D_{KL}}(p_0 || p_i) + \mathrm{D_{KL}}(p_0 || p_j) \overset{?}{=} \mathrm{D_{KL}}(p_0 || p_{i,j}), \tag{3}$$

where $p_i = p(x|\delta_i), p_0 = p(x|\delta_0)$ and $\mathrm{D_{KL}}(\cdot || \cdot)$ denotes the Kullback–Leibler (KL) divergence. Note that KL divergence measures the difference between two distributions; in our context, it quantifies the distribution shift resulting from interventions. Eq. (3) reflects the intuition that the influence of two separable perturbations, measured by KL divergence, is additive. We use the KL score, $|\mathrm{D_{KL}}(p || p_i) + \mathrm{D_{KL}}(p || p_j) - \mathrm{D_{KL}}(p || p_{i,j})|$, to quantify the violation of the separability of $\delta_i, \delta_j$, where these KL divergences are estimated using samples.

In practice, we observe that the simple K-nearest neighbor-based (KNN) estimator of the KL divergence (Wang et al., 2009) suffices for low-dimensional problems. However, for high-dimensional data such as images, more dedicated estimation procedures are required (Belghazi et al., 2018; Song & Ermon, 2020; Ghimire et al., 2021). We detail our estimation procedure in Appendix C.

### 3.2 DISJOINTEDNESS TESTING

Our second testing procedure examines whether two perturbations operate on disjoint domains such that their effects are cumulative. For example, if the morphological changes from knocking out two non-interacting genes can be separated into distinct visual features, such that their measures sum, then they are disjoint. We can define disjointedness of two perturbations formally as,

**Definition 3.7.** *Two perturbations $\delta_i, \delta_j$ are **disjoint** if*

$$p(x|\delta_{ij}) - p(x|\delta_0) = (p(x|\delta_i) - p(x|\delta_0)) + (p(x|\delta_j) - p(x|\delta_0)). \tag{4}$$

Disjointedness is important, because it implies that we can predict pairwise summary statistics of the distributions from individual perturbations. To see this, let $h(\cdot)$ denote *any* feature map from the observation, $X$. If we have disjoint perturbations $\delta_i, \delta_j$, then,

$$\mathbb{E}[h(x)|\delta_{i,j}] - \mathbb{E}[h(x)|\delta_0] = \int h(x)[p(x|\delta_{ij}) - p(x|\delta_0)]\mathrm{d}x$$

$$= \int h(x)[(p(x|\delta_i) - p(x|\delta_0)) + (p(x|\delta_j) - p(x|\delta_0))]\mathrm{d}x$$

$$= \mathbb{E}[h(x)|\delta_i] - \mathbb{E}[h(x)|\delta_0] + \mathbb{E}[h(x)|\delta_j] - \mathbb{E}[h(x)|\delta_0]. \tag{5}$$

This implies that we can define average centered embedding vectors, $\vec{h}_i := \mathbb{E}[h(x)|\delta_i] - \mathbb{E}[h(x)|\delta_0]$ and $\vec{h}_j$, and accurately predict $\vec{h}_{i,j} = \vec{h}_i + \vec{h}_j$ without running the experiments. Notably, growing

evidence in the literature (Lotfollahi et al., 2019; Gaudelet et al., 2024) showing that these relationships often hold in real biological experiments (and our experiments support this; see Appendix E.2 for the choice of $h$ we tested). Disjointedness explains the sufficient conditions for this to hold.

We can test whether Eq. (4) holds, by testing the following null hypothesis:

$$H_0 : \frac{1}{2}p(x|\delta_{ij}) + \frac{1}{2}p(x|\delta_0) = \frac{1}{2}p(x|\delta_i) + \frac{1}{2}p(x|\delta_j).$$

Given interventional data from $p(x|\delta_0), p(x|\delta_i), p(x|\delta_j)$ and $p(x|\delta_{ij})$, we can frame this as a standard two-sample test problem. We can test this using the maximal mean discrepancy (MMD) based two-sample test (Gretton et al., 2012), which compares two distributions based on their embeddings in some reproducing kernel Hilbert space (RKHS). The estimated MMD serves as a measure of the extent to which the perturbations violate the disjointedness. Interestingly, MMD test amounts to test Eq. (5) on a "most discriminative" feature map $h$ in the RKHS. We provide a self-contained introduction about MMD and kernel mean embedding of distributions in Appendix E. Disjointness is a constraint on the densities of the observed pixels, but it is implied by assuming an appropriate mixture model on the latent variables. We discuss this in Appendix D.

## 4 SELECTING PERTURBATION PAIRS TO EFFICIENTLY DISCOVER INTERACTIONS

With frameworks for testing separability and disjointedness in place, the next question is *how to select which experiments to run*. The space of possible perturbation pairs is too large to test all combinations, as experiments are costly. For instance, pairwise knockouts of 20,000 genes would require about 200 million experiments (not including replicates). In this section, we explore how experimental design and bandit algorithms can efficiently select perturbation pairs likely to interact.

**Selection of perturbation pairs as active matrix completion.**   The testing frameworks discussed in Section 3 prescribe test statistics which can be used to detect pairwise interactions. The test statistics, however, require samples from $p(x|\delta_{i,j})$, which entails running pairwise perturbation experiments. We are thus interested in developing an approach for selecting pairs of perturbations which are *likely* to have high values for the test statistics and are thus likely to reveal pairwise interactions.

The test statistics for pairs of perturbations can be viewed as an (unknown) symmetric matrix $\mathbf{R} \in \mathbb{R}^{n \times n}$, where each entry $\mathbf{R}_{i,j}$ contains the value of the test statistic that we will observe if we run perturbation $\delta_{ij}$. $\mathbf{R}$ is unknown a priori but we are allowed to select entries to observe. This can be viewed as an *active matrix completion* problem (Chakraborty et al., 2013). Active matrix completion is a variant of the standard matrix completion problem (Laurent, 2009) where values of entries of the matrix can be sequentially queried. Framing the problem of selecting perturbation pairs as an active matrix completion allows us to leverage existing efficient algorithms.

**Adaptive sampling for discovery.**   In particular, we use the framework of *adaptive sampling for discovery* (ASD; Xu et al., 2022) which provides a bandit-based approach for active matrix completion. ASD involves using *information directed sampling* (IDS; Russo & Van Roy, 2016) in a "discovery" setting, where an action is only selected once. This is an instantiation of the general sleeping experts setting (Kanade et al., 2009), where the set of available actions shrinks every round.

Denote $\Delta$ the action space of possible experiment designs, which in this case is the set of perturbations $\Delta := \{\delta_{ij} : i, j \in n, i > j\}$, where $n$ is the total number of distinct perturbations. We denote the perturbation pair $i, j$ selected at step $k$ as $a^{(k)} := (i^{(k)}, j^{(k)}) \in \Delta$. $\mathcal{D}(\Delta)$ represents the set of possible (categorical) distributions over $\Delta$. Each time an action $a^{(k)}$ is selected, the corresponding element of the (unknown) reward matrix $\mathbf{R}$ is revealed. In our setting, this reward corresponds to the test

---

**Algorithm 1** ASD for selecting perturbation pairs

1: Initialize $H_1 = \{\}$, batch size $b$
2: **for** $t = 1, \ldots, T$ **do**
3:     Estimate posterior $p(\mathbf{R} \mid H_t)$
4:     Compute information ratio $\Psi_t$
5:     Select $(a_i)_{i=1}^b$ greedily to maximize $\Psi_t$
6:     Perform experiments and obtain $(\mathbf{R}_{a_i})_{i=1}^b$
7:     Update $H_{t+1} = H_t \cup \{(a_i, \mathbf{R}_{a_i})\}_{i=1}^b$
8: **end for**

---

statistic from either Section 3.1 or 3.2 for each perturbation pair. Let $H_t = ((a^{(k)}, \mathbf{R}_{i^{(k)}, j^{(k)}}))_{k=1}^{t-1}$ denote the history of actions and their corresponding rewards until round $t$, and $\Delta_t$ the set of remaining actions; i.e. the perturbation pairs that we have not yet tested experimentally. A policy $\pi$ is a map from $H_t$ to $\mathcal{D}(\Delta_t)$. The IDS policy, $\pi_{\text{IDS}}$ maintains a posterior distribution over $\mathbf{R}$ given the data observed up to round $t$, denoted $p(\mathbf{R} \mid H_t)$.

We can describe the sub-optimality of any action with respect to a set of beliefs by comparing the action's reward to that of the best action that could have been selected at time $t$, under the agent's current posterior over the reward matrix. Intuitively, we can evaluate this by sampling a plausible reward matrix from our posterior, $\dot{\mathbf{R}} \sim (\mathbf{R}|H_t)$, and then comparing the reward from action, $a$, to the reward an agent would have received from selecting the optimal action, $a^* = \arg\max_{a \in \Delta_t} \dot{\mathbf{R}}(a)$; where $\dot{\mathbf{R}}(a^{(k)}) := \dot{\mathbf{R}}_{i^{(k)}, j^{(k)}}$ for $a^{(k)} = (i^{(k)}, j^{(k)})$. This is known as the expected instantaneous regret incurred by an action and is defined as, $\Delta_t(a) = \mathbb{E}_{\dot{\mathbf{R}} \sim p(\mathbf{R}|H_t)}[\dot{\mathbf{R}}(a^*) - \dot{\mathbf{R}}(a)]$. Additionally, we define the information gain about the top $T - t + 1$ remaining actions as follows:

$$g(a) = MI(a^*_{t,1}, \ldots, a^*_{t,T-t+1}; \mathbf{R}(a) \mid H_t, \delta_t = a).$$

Algorithm 1 summarizes the algorithmic procedure. The algorithm operates over a series of $T$ rounds. In each round, the first step is to estimate the posterior distribution over $\mathbf{R}$ given the data observed thus far. Next step involves selecting a batch of perturbations based on the information ratio. The IDS policy at round $t$ can then be computed by minimizing the *information ratio* $\Psi$:

$$\pi_{\text{IDS}} \in \arg\min_{\pi \in \mathcal{D}(\Delta_t)} \Psi_{\pi,t} := \frac{(\Delta_t^\top \pi)^\lambda}{g_t^\top \pi}$$

where $\lambda$ is a parameter controls the tradeoff between lower instant regret (exploitation) and higher information gain (exploration). Due to the intractability of $g_t(a)$, we follow Russo & Van Roy (2016); Xu et al. (2022) to approximate $g_t$ with the conditional variance $v_t(a) = \text{Var}_t(\mathbb{E}[\mathbf{R}_a|a^*_1, a])$. $v_t(a)$ is a lower bound on the information gain $g_t(a)$ and hence can be used as a surrogate when maximizing $g_t(a)$. After selecting the perturbation pair $i, j$, we compute the pairwise test statistic, $\mathbf{R}_{i,j}$ using data from the experimental outcomes. We add these test statistics to our reward matrix, update our posteriors and then continue on to the next round. Linear regret is unavoidable for any bandit algorithm with no structural assumptions on $\mathbf{R}$. We thus assume $\mathbf{R}$ is low-rank with a Gaussian prior on the entries, allowing ASD to achieve sublinear regret (Xu et al., 2022). In our motivating biological example the low-rank assumption amounts to assume that there are morphological similarities in the functional response of cells to genetic perturbations (e.g. because knockouts of different genes on the same pathway product the same morphological effect). We validate this empirically in Figure 5 and discuss the assumption further in subsection F.3.

**Batching.** In high-throughput experimental screens, it is possible to run multiple experiments in parallel at the cost of a single experiment. So instead of a single action $\delta_t$, we can select a set of actions $\{\delta_t^1, \ldots, \delta_t^b\}$ where $b$ is the number of experiments we can run in parallel. However, as there are no efficient algorithms for combinatorial bandits in the discovery setting, we resort to a simple greedy scheme to select batches with ASD. Specifically, instead of picking a single action which minimizes the information ratio, we pick $b$ actions with the lowest information ratio.

## 5 EXPERIMENTS

Our experiments aim to address three objectives: (1) verifying that our theoretical claims about interactions are detectable on known synthetic tasks; (2) evaluating the test statistics' ability to recover known biological relationships from real pairwise perturbation experiments; and (3) assessing our active learning pipeline's efficiency in detecting interactions. All experimental details are provided in Appendix F.

### 5.1 TESTING ON SYNTHETIC SETTING

We validated the separability test on two synthetic interventional examples: one with 3-dimensional tabular data and one with images ($3 \times 128 \times 128$). The disjointedness test was validated on an interventional tabular example. In each case, we generated observations by sampling from a latent distribution $p_Z$ following a DAG structure, followed by a mapping $g(\cdot)$ to the observations. We then estimated the test statistics for each pair of perturbations. Detailed descriptions of the data generation processes, DAG structures, and mappings are provided in Appendix F.1.

Fig. 2 displays the separability test results for both tabular and image data, showing that our estimated KL scores accurately capture separability: inseparable pairs have large KL scores, while separable pairs have small scores. Fig. 4 shows the disjointedness tests, with the MMD test accurately identifying failures of disjointedness and being relatively insensitive to the choice of kernel.

### 5.2 TESTING ON BIOLOGICAL INTERACTIONS

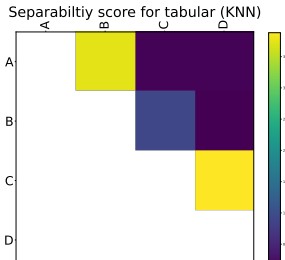
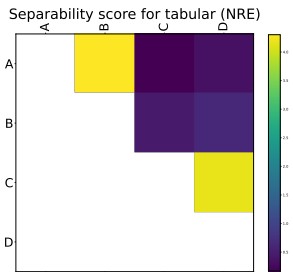
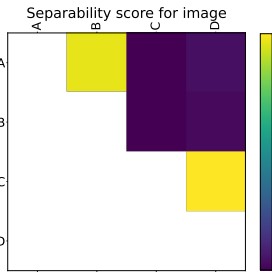

Figure 2: Separability test on synthetic tabular data using the KNN-based KL estimator (*left*) and NRE-based KL estimator (*middle*), and on synthetic images (*right*). Brighter colors indicate stronger interactions. Ground truth interacting pairs, A-B and C-D, are correctly identified in both examples.

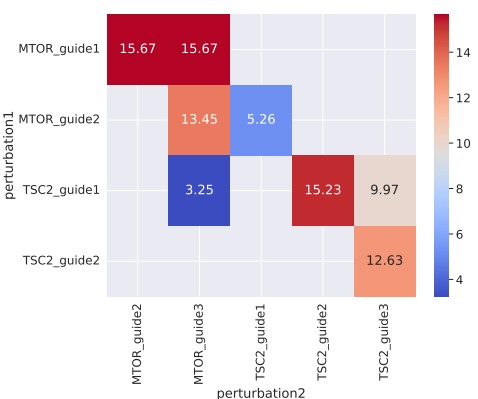
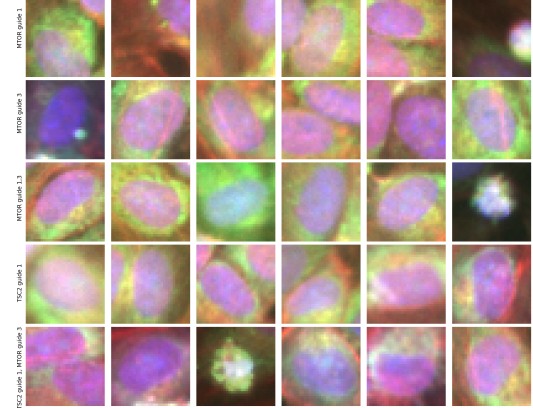

Figure 3: *(Left)* Pairwise separability scores between different CRISPR guides targeting the two genes TSC2 and MTOR. Missing pairs indicate that data for those combinations were not collected in our experiments. Guides targeting the same gene show high KL scores (red), while guides targeting different genes show low scores (blue). *(Right)* Random samples of the actual single-cell images used in these experiments. Even trained experts find it very difficult to detect interactions from the image.

To evaluate if our separability test can recover known biological interactions, we ran the following test: In gene knockout experiments, different CRISPR guides can target the same gene, cutting it at different locations but resulting in the same gene knockout. Intuitively, guides targeting the same gene should show high separability scores, while guides targeting different genes should show lower scores, assuming the genes are on distinct pathways. While it's rare for different guides targeting the same gene to be run in a single well, our dataset includes examples for two genes (TSC2 and MTOR). Using single-cell painting images, we tested separability between pairs of guides. The results, shown in Fig. 3, align with expectations: strong interaction scores were observed between guides targeting the same gene (e.g., MTOR guide 3 with MTOR guides 1 and 2), and much weaker scores between MTOR and TSC2 guides. Both MTOR and TSC2 impact multiple cell systems, so some interaction is expected, but the significantly lower scores between genes compared to within the same gene are encouraging.

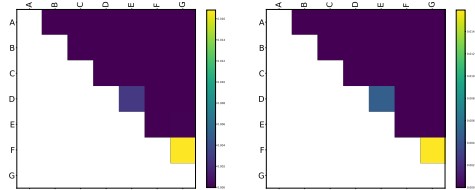

Figure 4: Disjointedness testing on synthetic example using the MMD-based statistics with a Matern 2.5 kernel (*left*) and an RBF kernel (*right*); brighter colors suggest stronger interactions. The ground truth interacting pairs are D-E and F-G, which are correctly identified by the test.

We then evaluated our testing approaches on a collection of 50 genes with the goal of detecting gene interactions. The dataset was collected by performing CRISPR knockouts on all pairs from a set of 50 selected genes, resulting in 1,225 gene pairs. The targeted genes have a bias towards known gene-gene interactions. We performed *in vitro* double gene knockout experiments on HUVEC cells using three

CRISPR guides per gene. We label perturbations according to the targeted gene, aggregating the effects of the individual CRISPR guides. The experimental protocol and data preprocessing followed the procedures described by Fay et al. (2023) and Sypetkowski et al. (2023), respectively.

We computed the test statistics using 1024-dimensional embeddings of the original cell painting images extracted from a pre-trained masked autoencoder (He et al., 2022; Kraus et al., 2023). Fig. 5 shows the matrices of the test statistics for all perturbation pairs; each entry represents the pairwise interaction scores. Qualitatively, both the disjointedness and separability statistics effectively uncover plausible biological relationships. Many genes in the apoptosis pathway (programmed cell death, e.g., gene 2–3: BAX, BCL2L1) and the proteasome (protein degradation e.g., gene 28–30: PSMA1, PSMB2, PSMD1) show high scores, expected from synthetic lethal (SL) relationships. SL is a complex phenomenon in which simultaneous inactivation of specific gene combinations leads to cell death or extreme sickness, while individual perturbations have little effect. Apoptosis, controlling cell survival, is tightly regulated; disruption of the anti-apoptotic BCL-2 family (BCL2, BCL2L1, MCL1) (Kale et al., 2018) interferes with critical barriers against cell death. Proteasome function is similarly regulated, as cells must maintain a delicate balance of different proteins (Rousseau & Bertolotti, 2018). Apoptosis and proteasome members are commonly found in SL screens in cancer cells due to mutations that compromise cellular buffering capabilities (Li et al., 2020; Ge et al., 2024; Han et al., 2017; Cron et al., 2013; Steckel et al., 2012; Das et al., 2020).

As shown in Fig. 5, the MMD-based statistics reveal strong interaction signals with proteasome components, which regulate global protein levels and interact with multiple essential pathways. While the separability score shows less distinct patterns, it highlights several gene pairs known or expected to interact physically or genetically, such as BCL2L1-MCL1 (genes 3-8) (Shang et al., 2020; Carter et al., 2023), BAX-BCL2L1 (genes 2-3) (Lindqvist & Vaux, 2014), BCL2L1-PSMD1 (genes 3-30) (Craxton et al., 2012), and PSMB2-PSMD1 (genes 29-30) (Voutsadakis, 2017).

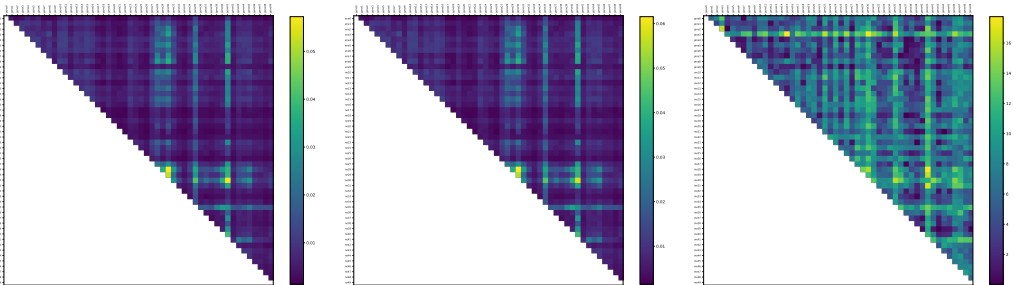

Figure 5: Pairwise interaction scores from a selected set of 50 genes using disjointedness test with Matern 2.5 kernel and RBF kernel (*left* and *middle*, respectively), and separability test (*right*); brighter colors suggest stronger interactions. Genes from the same pathway are ordered adjacently. The associated pathways of each selected gene are described in Table 1 of Appendix F.2.1.

## 5.3 Interaction discovery with active matrix completion

Finally, we showcase how to use our test statistics to adaptively select the pairwise experiment to run. As introduced in Section 4, the adaptive experimental design is framed as an active matrix completion problem (Algorithm 1). We applied Algorithm 1 on the gene-gene interaction detection example using the MMD-based scores.

**Baselines.** To evaluate the effectiveness of our automated selection method, we compare against selection with random policy, upper confidence bound (UCB), Thompson sampling (TS) and uncertainty sampling (US). To instantiate UCB in the discovery setting, we use the uncertainty from the low-rank matrix posterior in place of the counts used in the standard multi-armed bandit setting and mask the actions once they are selected. Similarly, in US we pick pairs based solely on the uncertainty of the posterior. TS is instantiated in the same way as IDS, but the pairs are selected to minimize only the instant regret. Further details are discussed in Appendix F.

**Evaluation metrics.** We evaluate each approach using three different metrics given a budget of 50 rounds, each with a batch size of 10 resulting in a total of 500 experiments covering 50% of all

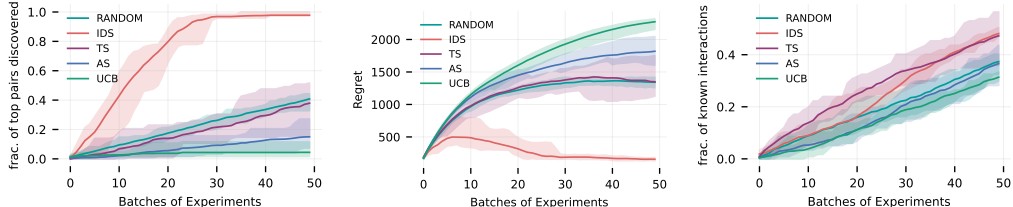

Figure 6: Empirical results on the active pairwise experiments selection for the gene-gene interaction detection example using MMD based test statistics. The solid lines represent the mean performance, whereas the shaded region represent all the runs (min-max). IDS (ours) outperforms all the baselines significantly in terms of top scoring pairs discovered as well as the regret. In terms of known interactions, IDS still outperforms the baselines by a small margin.

possible pairs of the gene set. First, we access the fraction of the pairs with the top 5 percentile of scores recovered by each algorithm, measuring how well they explore high-scoring regions. Next, we evaluate the regret of each algorithm with respect to an optimal policy that always selects the highest-scoring pairs. Finally, we evaluate the number of known biological relations that appear in CORUM (Giurgiu et al., 2019), StringDB (Szklarczyk et al., 2021), Signor (Licata et al., 2020) and hu.MAP (Drew et al., 2021) to see how many each method is able to recover.

**Results.** Fig. 6 (and Fig. 8 in Appendix F.3) show the empirical results. IDS successfully discovers all pairs in the top 5 percentile, while the baselines recover only about half. In terms of regret, IDS outperforms the baselines, with random performing the worst, demonstrating IDS's ability to exploit the low-rank structure of the reward matrix. Regarding known interactions discovered, IDS and TS slightly outperform other methods. After 50 rounds, IDS and TS show a $12 - 15\%$ improvement over the baselines in identifying known biological interactions. The significant difference between the performance on the fraction of top pairs and the number of known relations raises questions about the correlation between the prediction errors and the biological interactions.

# 6 DISCUSSION

This paper presented a method for efficiently detecting interactions. We present interactivity scores for both separability and disjointedness which both allow us to measure interactions between perturbations, and run experiments only on pairs of perturbations for which we will fail to compositionally generalize. From an active learning perspective, disjointedness is powerful: when we can learn a good posterior over this test statistic, we effectively have a "confidence score" for whether two perturbations are likely to compose additively. This allows us to dramatically reduce the number of experiments we run by experimenting only where embeddings are unpredictable. Separability is a more intuitive notion of (in-)dependence in that we directly test for whether perturbations interact in latent space; we aim to explore whether this can be used to recover intervention targets.

**Limitations** While many known interactions scored highly according to our test statistics, the overall correlation between known interactions and this metric was relatively low. It is not clear whether this is the result of a lack of specificity, or whether this is because we are in fact discovering real relationships that are not known to biology. To test this would require additional experimentation with orthogonal assays. Additionally, the results for the separability tests depended on the quality of the KL estimator. We used the SMILE estimator which is somewhat sensitive to the choice of the clipping parameter, $\tau$. In our experiments, we simply used a default of $\tau = 5$, but we need a robust method for choosing this hyperparameter to ensure this test is agnostic to modalities. Finally, the separability score test lacks a rigorous rejection criterion, and a characterization of type I and type II error. While this is less critical when the goal is to use the test statistic for active learning (introduced later in Section 4), a formal hypothesis testing framework is important from a statistical perspective. This issue has been addressed in a follow-up work.

## REPRODUCIBILITY STATEMENT

We provide the proofs for all our theoretical results along with clearly stated assumptions in Appendix B. We describe all the details and provide the hyperparameters used for our experiments in Appendix F. We also provide code for the synthetic experiments here. The experiments on biological data rely on proprietary data and models which cannot but released, but we do our best to describe the exact experimental setup.

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

# Appendix

## Table of Contents

## A    RELATED WORK ON INTERACTION DETECTION

**Biological interactions.**    Ahlmann-Eltze et al. (2024) is concurrent work that explores the feasibility of predicting measurements from double gene knockouts based on single gene knockouts. They show empirically that for certain gene pairs, the simple additive relationship $\mathbb{E}[x|\delta_{i,j}] = \mathbb{E}[x|\delta_i] + \mathbb{E}[x|\delta_j] - \mathbb{E}[x|\delta_0]$ holds for many pairs $i, j$, where $x$ denotes the measurements of gene expression. This model is a special case of Eq. (5) when the embedding function $h(\cdot)$ is the identity function. Consequently, definition 4 can be view as a sufficient condition for when the additive model proposed in Ahlmann-Eltze et al. (2024) provides accurate predictions for double knockouts and our disjointedness test can be seen as a method for testing whether this holds. Our approach, however, is more general, as it establishes sufficient conditions for additivity under any embedding function $h(\cdot)$. Rather than predict double perturbation outcomes directly—which, as noted by Ahlmann-Eltze et al. (2024), has seen limited success—we focus on identifying cases where the additive assumption fails so that we can measure these outcomes experimentally. The heat maps in Fig. 5 (left and middle) highlight two different estimates of where this failure occurs. Consistent with Ahlmann-Eltze et al. (2024), there are many regions where the additive model holds (dark blue areas) but also significant regions where it breaks down. Akimov & Aittokallio (2021) examined the potential benefits of leveraging high-throughput functional and genomic screening to discover novel therapeutically relevant synthetic lethality (SL) interactions. They emphasize the importance of expanding the definition of synthetic lethality to include broader notions such as "synthetic sickness." However, they do not provide methods for detecting these more general biological interactions. This gap serves as a strong motivation for our work. We define two broad notions of interaction, rigorously characterized by probabilistic models, and propose general inference procedures to efficiently detect these interactions.

**Statistical interactions of treatments.**    Statistical interaction tests for multiple treatments or covariates (Cox, 1984; de González & Cox, 2007) are traditionally studied within the framework of

classical statistical design of experiments using factor analysis and regression models. These methods primarily focus on scalar outcome variables, modeling relationships between the response and factors (treatments) through linear or generalized linear regressions with explicit parametric assumptions. Interactions are interpreted as non-additive effects from multiple treatments (a special instance of Eq. (5)), with analysis of variance (ANOVA) or likelihood ratio tests employed to assess the statistical significance of such non-additivity. In contrast, our work aims to generalize the notion of interaction to unstructured outcome data, such as image pixels. This task is significantly more challenging due to the high dimensionality of the data and the need for our notion of interaction to remain robust to arbitrary bijective transformations (since we cannot assume control over the measurement process). As de González & Cox (2007) point out, classical interaction testing approaches are highly sensitive to transformations, where a simple transformation of the response variable, such as $y$ to $y^{1/3}$, can change the statistical conclusion of the presence of interactions. Similar interaction screening techniques extend to regression models with multivariate responses (Kong et al., 2017; Xinyi Li, 2021), but sill, these methods are tied to specific (generalized) linear models and do not offer the generality or robustness of our framework.

# B PROOFS

*Proof of Theorem 3.6.* By Assumptions 3.4 and 3.2, we can apply change of variable formula to obtain the log density of $p_X(x|T_1, \ldots, T_n)$:

$$\log p(x|T_1, \ldots, T_n) = \sum_{l=1}^{L} \log p_{z_l}([g^{-1}(x)]_l | \operatorname{Pa}(z_l)) + \log |\det \nabla g^{-1}(x)|.$$

Here $[\cdot]_l$ denote the projection operator that maps $z \in \mathcal{Z}$ to the subspace on which $z_l$ lies, i.e., $\forall z \in \mathcal{Z}, [z]_l = z_l$. Recall that we interpret $z$ as the concatenation of all latent factors $(z_1, \ldots, z_L)$. If $T_i, T_j$ are causally independent, perturbation $\delta_i, \delta_j$ will intervene different terms in the above summand. Without lose of generality, suppose $\delta_i$ and $\delta_j$ intervene $z_{l_i}$ and $z_{l_j}$ repsectively. Then,

$$\bar{\ell}_X(x|\delta_i) = \log p(x|\delta_i) - \log p(x|\delta_0)$$
$$= \log p_{z_{l_i}}([g^{-1}(x)]_{l_i}|\delta_i) - \log p_{z_{l_i}}([g^{-1}(x)]_{l_i}|\delta_0).$$

where the equality follows from the modularity assumption that the intervention only affects $l_i$.

Similarly, we obtain that

$$\bar{\ell}_X(x|\delta_j) = \log p_{z_{l_j}}([g^{-1}(x)]_{l_j}|\delta_j) - \log p_{z_{l_j}}([g^{-1}(x)]_{l_j}|\delta_0),$$
$$\bar{\ell}_X(x|\delta_{ij}) = \log p_{z_{l_i}}([g^{-1}(x)]_{l_i}|\delta_i) - \log p_{z_{l_i}}([g^{-1}(x)]_{l_i}|\delta_0)$$
$$+ \log p_{z_{l_j}}([g^{-1}(x)]_{l_j}|\delta_j) - \log p_{z_{l_j}}([g^{-1}(x)]_{l_j}|\delta_0),$$

which completes the proof. $\qquad\square$

*Proof of Theorem D.2.* Provided with Assumption 3.1, $\delta_i, \delta_j$ being disjoint (Definition 3.7) is equivalent to the same additivity in the intervened latent distributions, i.e.,

$$p_Z(z|\delta_{ij}) - p_Z(z|\delta_0) = p_Z(z|\delta_i) - p_Z(z|\delta_0) + p_Z(z|\delta_j) - p_Z(z|\delta_0). \quad (6)$$

Under the mixture model assumption Assumption D.1, if $T_i, T_j$ are causally independent, if $\operatorname{Ch}(T_i) \cap \operatorname{Ch}(T_j) = \emptyset$, perturbation $\delta_i, \delta_j$ will intervene distinct mixing components. Without lose of generality, suppose $\delta_i$ and $\delta_j$ intervene $p_{Z_{l_i}}$ and $p_{Z_{l_j}}$ repsectively.

$$p_Z(z|\delta_i) - p_Z(z|\delta_0) = w_{l_i} \cdot (p_{Z_{l_i}}(z|\delta_i) - p_{Z_{l_i}}(z|\delta_0)),$$
$$p_Z(z|\delta_j) - p_Z(z|\delta_0) = w_{l_j} \cdot (p_{Z_{l_j}}(z|\delta_j) - p_{Z_{l_j}}(z|\delta_0)),$$
$$p_Z(z|\delta_{ij}) - p_Z(z|\delta_0) = w_{l_i} \cdot (p_{Z_{l_i}}(z|\delta_i) - p_{Z_{l_i}}(z|\delta_0)) + w_{l_j} \cdot (p_{Z_{l_j}}(z|\delta_j) - p_{Z_{l_j}}(z|\delta_0)),$$

which shows Eq. (6) and hence completes the proof. $\qquad\square$

## C  SAMPLE-BASED ESTIMATION OF KL DIVERGENCE

Sample-based estimator of KL-divergence is a challenging task, particularly for high dimensional observations. In this section, we present the estimation procedure we used in our experiments. As mentioned in Section 5, to estimate the KL-divergence,

$$D_{KL}(p||p_i) = \mathbb{E}\left[\log \frac{p(X|\delta_0)}{p(X|\delta_i)}\right], \quad X \sim p(X|\delta_0), \tag{7}$$

we used a two-step procedure:

1. We first estimate the log-density ratio

$$\log \frac{p(x|\delta_0)}{p(x|\delta_i)} \tag{8}$$

   via a neural ratio estimator (NRE) based on a contrastive learning objective (Hermans et al., 2020, NRE).

2. Then, instead of taking naive Monte Carlo estimates of the KL-divergence, we adopt the SMILE estimator (Song & Ermon, 2020) using the learned log-density ratio.

Appendices C.1 and C.2 describe the two steps respectively.

### C.1  CONTRASTIVE NEURAL RATIO ESTIMATOR

There are various methods to learn the log-density ratio between a single pair of distributions. For example, the optimal discriminator between two distributions is related to their density ratios (Goodfellow et al., 2014, Proposition 1). However, a common structure in our applications involves a large set of perturbations, and training a classifier for each perturbation class (against the control group) would be very cumbersome and not data efficient. Therefore, we consider a contrastive learning model (Hermans et al., 2020, NRE), which trains a binary classifier to distinguish the joint data distribution $p(x, c)$ from the product of the marginals $p(x)p(c)$, where $c$ denotes the perturbation class.

The training objective is as follows:

$$\theta, w \in \arg\min_{\theta, w} -\frac{1}{2B}\left[\sum_{b=1}^{B}\log(1 - \text{Sigmoid}(f_{\theta,w}(x^{(b)}, c^{(b)}))) + \sum_{b'=1}^{B}\log(\text{Sigmoid}(f_{\theta,w}(x^{(b')}, c^{(b')})))\right],$$

where $x^{(b)}, c^{(b)} \sim p(x)p(c)$ and $x^{(b')}, c^{(b')} \sim p(x, c)$, and

$$f_{\theta,W}(x, c) = \text{Encoder}_\theta(x)^T W_c.$$

Given infinite training samples and flexibility of the neural network $f$, the optimal $f^\star(x, c) = \log \frac{p(x,c)}{p(x)p(c)} = \log \frac{p(x|c)}{p(x)}$. And we can then obtain Eq. (8) via $f^\star(x, \delta_i) - f^\star(x, \delta_0)$.

In our examples, we estimate our log-density ratios by training the NRE objective on all perturbation classes.

### C.2  SMOOTHED MUTUAL INFORMATION "LOWER-BOUND" ESTIMATOR (SMILE)

After obtaining the log-density ratio estimator for Eq. (8), there are several options for estimating the KL-divergence. We provide a short review here on the various strategies (Ghimire et al., 2021; Belghazi et al., 2018; Hjelm et al., 2019; Song & Ermon, 2020), and explain why we opt for the SMILE estimator (Song & Ermon, 2020).

The most straightforward estimates of the KL-divergence Eq. (7) is by the Monte Carlo estimates based on samples from $p(X|\delta_0)$, i.e.,

$$D_{KL}(P(X|\delta_i)||P(X|\delta_0)) \approx \frac{1}{N}\sum_{i=1}^{N} f(X_i), \quad \text{where } f(\cdot) \text{is the estimated} \log \frac{p(\cdot|\delta_i)}{p(\cdot|\delta_0)}.$$

However, it is noted that the variance of this estimator is often huge (Song & Ermon, 2020; Ghimire et al., 2021), making the estimated KL unreliable in practice.

Belghazi et al. (2018) proposed to estimate the KL-divergence based on its Donsker-Varadhan representation (Donsker & Varadhan, 1983), i.e.,

$$D_{KL}(p||q) = \sup_{f:\Omega \to \mathbb{R}} \mathbb{E}_p[f] - \log \mathbb{E}_q[\exp(f)],$$

where the supremum is taken over all functions $f$ such that the two expectations are finite. Notice that the optimal $f$ is indeed achieved as the log-density ratio between $p$ and $q$. In practice, one can parameterize $f$ using some neural network and maximize the above objective to approximate the KL (Belghazi et al., 2018). However, the stochastic gradient estimator of the above objective is generally biased (Belghazi et al., 2018), making the optimization less stable. Therefore, it is also recommended to first learn the log-density ratio $\log \frac{p}{q}$ as $f$, and then estimate the KL-divergence (Hjelm et al., 2019; Song & Ermon, 2020) via

$$KL_{MILE}(f) = \mathbb{E}_p[f] - \log \mathbb{E}_q[\exp(f)],$$

which technically gives a lower-bound of the KL divergence if the log-density ratio is not well-estimated. We would refer this estimator as the mutual information lower-bound estimator (MILE). MILE is often shown to have better performance than the naive Monte Carlo estimator (Belghazi et al., 2018; Hjelm et al., 2019; Song & Ermon, 2020).

However, MILE also suffers from high variance issues (Song & Ermon, 2020) particularly when the learned $f$ has large values in the tail of $q$. Song & Ermon (2020) proposed a smoothed version of MILE, named as SMILE, by clipping the learned log-density ratios $f$ between $-\tau$ and $\tau$, i.e.,

$$KL_{SMILE}(f, \tau) = \mathbb{E}_p[f] - \log \mathbb{E}_q[\text{clip}(\exp(f), \exp(-\tau), \exp(\tau))].$$

$KL_{SMILE}(f, \tau)$ converges to $KL_{MILE}(f)$ as $\tau \to \infty$, but smaller $\tau$ significantly reduces the variance of the MILE estimator.

In our experiments, we use the SMILE estimator with the clipping parameter $\tau$ set to be $5$.

## D  MIXTURE EXAMPLE

Unlike the separability formulation, which assumes the existence of the Markov factorization of the latent distribution (Assumption 3.4), the disjointedness models the latent distribution as a finite mixture. In this framework, non-interacting perturbations intervene on different mixing components.

**Assumption D.1.** *The latent distribution $p_Z(z)$ admits the form of an $L$-component mixture:*

$$p_Z(z|T_1, \ldots, T_n) = \sum_{l=1}^{L} w_l \cdot p_{Z_l}(z|\text{Pa}(z_l)), \quad \text{where } (w_1, \ldots, w_L) \in \Delta_L.$$

*In addition, perturbations do not intervene the mixing weights $(w_1, \ldots, w_L)$.*

**Theorem D.2.** *Under Assumptions 3.1 and D.1, $\delta_i, \delta_j$ are disjoint if $\text{Ch}(T_i) \cap \text{Ch}(T_j) = \emptyset$.*

## E  MMD AND KERNEL MEAN EMBEDDING

### E.1  KERNEL MEAN EMBEDDING

We provide here a minimal overview of the kernel mean embedding of probability measures (Muandet et al., 2017). Given a feature map $\phi : \mathcal{X} \to \mathcal{F}_\phi$, where $\mathcal{F}_\phi$ is some Hilbert space (sometimes called the feature space). This feature map $\phi$ defines a kernel:

$$k_\phi : \mathcal{X} \times \mathcal{X} \to \mathbb{R}, \qquad k_\phi(x, y) = \langle \phi(x), \phi(y) \rangle,$$

where $\langle \cdot, \cdot \rangle$ denotes the inner product of $\mathcal{F}_\phi$. Kernel $k_\phi$ defined this way induces a space of functions—$\mathcal{H}_\phi$—from $\mathcal{X}$ to $\mathbb{R}$, which is a *reproducing kernel Hilbert space* (RKHS). The name RKHS comes from a special property of $\mathcal{H}_\phi$, called the *producing property*:

$$\forall f \in \mathcal{H}_\phi, \qquad \langle f, \phi(x) \rangle_{\mathcal{H}_\phi} = f(x).$$

We abuse the notation here by interpreting $\phi(x)$ as a function in $\mathcal{H}_\phi$ instead of a real value. A special instance of the reproducing property is that

$$\langle \phi(x), \phi(y) \rangle = k_\phi(x, y).$$

Precisely, we are using the following representation of $\phi(x)$:

$$\phi(x) : \mathcal{X} \to \mathcal{H}_\phi, \qquad \phi(x)(\cdot) = k_\phi(x, \cdot) \in \mathcal{H}_\phi.$$

Here $k_\phi(x, \cdot) \in \mathcal{H}_\phi$ is guaranteed by the definition of $\mathcal{H}_\phi$.

In what follows, we use $\mathcal{M}_+^1(\mathcal{X})$ to denote the space of probability measures over $\mathcal{X}$. We can embed any probability measure to the RKHS.

**Definition E.1** (Kernel mean embedding of probability measure). *The* kernel mean embedding *of a probability measure $\mathbb{P} \in \mathcal{H}_\phi$ is defined via the following mapping:*

$$\mu_\phi : \mathcal{M}_+^1(\mathcal{X}) \to \mathcal{H}_\phi, \qquad \mu_\phi(\mathbb{P})(\cdot) = \int_\mathcal{X} \phi(x)(\cdot)\mathbb{P}(\mathrm{d}x) = \mathbb{E}_{X \sim \mathbb{P}}[\phi(X)](\cdot).$$

We denote the kernel mean embedding of $\mathbb{P}$ by $\mu_\phi(\mathbb{P})$.

**Proposition E.2** (c.f Eq. (3.29) of Muandet et al. (2017)). *For all $\mathbb{P}, \mathbb{Q} \in \mathcal{M}_+^1(\mathcal{X})$,*

$$\mathrm{MMD}_{k_\phi}(\mathbb{P}, \mathbb{Q}) = \|\mathbb{E}_{X \sim \mathbb{P}}[\phi(X)] - \mathbb{E}_{Y \sim \mathbb{Q}}[\phi(Y)]\| = \|\mu_\phi(\mathbb{P}) - \mu_\phi(\mathbb{Q})\|. \tag{9}$$

*Proof of Proposition E.2.* The second equality of Eq. (9) follows directly from the mapping defined in Definition E.1. We then focus on proving the first equality.

$$\mathrm{MMD}_{k_\phi}(\mathbb{M}_1, \mathbb{M}_2)$$
$$= \sup_{f \in \mathcal{H}_\phi : \|f\| \leq 1} \mathbb{E}_{X \sim \mathbb{P}}[f(X)] - \mathbb{E}_{Y \sim \mathbb{Q}}[f(Y)]$$
$$= \sup_{f \in \mathcal{H}_\phi : \|f\| \leq 1} \mathbb{E}_{X \sim \mathbb{P}}[\langle f, \phi(X)\rangle] - \mathbb{E}_{Y \sim \mathbb{Q}}[\langle f, \phi(Y)\rangle] \quad \text{(reproducing property)}$$
$$= \sup_{f \in \mathcal{H}_\phi : \|f\| \leq 1} \langle f, \mathbb{E}_{X \sim \mathbb{P}}[\phi(X)]\rangle - \langle f, \mathbb{E}_{Y \sim \mathbb{Q}}[\phi(Y)]\rangle \quad \text{(linearity of inner product)}$$
$$= \sup_{f \in \mathcal{H}_\phi : \|f\| \leq 1} \langle f, \mathbb{E}_{X \sim \mathbb{P}}[\phi(X)] - \mathbb{E}_{Y \sim \mathbb{Q}}[\phi(Y)]\rangle \quad \text{(linearity of inner product)}$$

Then the proof is completed by the observation that the inner product is maximized when

$$f = \{\mathbb{E}_{X \sim \mathbb{P}}[\phi(X)] - \mathbb{E}_{Y \sim \mathbb{Q}}[\phi(Y)]\} / \|\mathbb{E}_{X \sim \mathbb{P}}[\phi(X)] - \mathbb{E}_{Y \sim \mathbb{Q}}[\phi(Y)]\|.$$

$\square$

Proposition E.2 essentially says that MMD between two distributions is indeed the distance of mean embeddings of features. It also tells us that $\mathrm{MMD}_{k_\phi}(\mathbb{P}, \mathbb{Q}) = 0$ if and only if $\mu_\phi(\mathbb{P}) = \mu_\phi(\mathbb{Q})$. To be able to separate any two distributions via MMD, the kernel mean embedding must be an injective map, in which case the feature map induces a *characteristic kernel*.

**Definition E.3** (Characteristic kernel). *$k$ is said to be characteristic on $\mathcal{M}_+^1(\mathcal{X})$ if the kernel mean embedding,*

$$\mu_k : \mathcal{M}_+^1(\mathcal{X}) \to \mathcal{H}_k, \qquad \mu_k(\mathbb{P})(\cdot) = \int_\mathcal{X} \phi(x)(\cdot)\mathbb{P}(\mathrm{d}x),$$

*is injective. In other words, $k$ is a characteristic kernel if and only if*

$$\mu_k(\mathbb{P}, \mathbb{Q}) = 0 \iff \mu_k(\mathbb{P}) = \mu_k(\mathbb{Q}) \iff \mathbb{P} = \mathbb{Q}. \tag{10}$$

Here we change the subscript of $\mu, \mathcal{H}$ from the feature map to the kernel, because from now on we do not necessarily work on explicit choice of feature maps; many characteristic kernels do not have tractable feature maps (and mostly infinite dimensional). Eq. (10) states that MMD is a metric on $\mathcal{M}_+^1(\mathcal{X})$ if a characteristic kernel is used; otherwise, distinct distributions with the same kernel mean embedding cannot be separated by MMD. Examples of characteristic kernels on $\mathcal{M}_+^1(\mathbb{R}^d)$ are Gaussian kernels, Laplacian kernels, and the family of Matérn kernels. We refer readers to Sriperumbudur et al. (2011) for a comprehensive survey of the characteristic kernels.

Provided with a specified kernel, and samples from $\mathbb{P}, \mathbb{Q}$, one can obtain an unbiased estimate of the squared population MMD; see Gretton et al. (2012, Eq. (3)) for the detailed expression of the estimator. An interesting property of the estimates of MMD is that the convergence is dimension independent (Gretton et al., 2012, Theorem 7), although the penalization of the dimension in the context of kernel two sample test lies in the reduction of power (Ramdas et al., 2015).

### E.2 COMPARE DISTRIBUTIONS VIA A FIXED FEATURE MAP $h$

In some cases, one can compare $\mathbb{P}, \mathbb{Q}$ by assessing on a fixed test function $h$. For example, the L2 norm between the expectation of $h$, i.e.,

$$\|\mathbb{E}_{X \sim \mathbb{P}}[h(X)] - \mathbb{E}_{Y \sim \mathbb{Q}}[h(Y)]\|_2 \tag{11}$$

can be a crude measure on how different $\mathbb{P}, \mathbb{Q}$. According to Proposition E.2,

$$\|\mathbb{E}_{X \sim \mathbb{P}}[h(X)] - \mathbb{E}_{Y \sim \mathbb{Q}}[h(Y)]\|_2 = \mathrm{MMD}_{k_h}(\mathbb{P}, \mathbb{Q}).$$

The obvious limitation with a fixed choice of $h$ is that it's typically not characteristic for all distributions, i.e.,

$$\|\mathbb{E}_{X \sim \mathbb{P}}[h(X)] - \mathbb{E}_{Y \sim \mathbb{Q}}[h(Y)]\|_2 = 0 \implies \mathbb{P} = \mathbb{Q}.$$

However, in practice, if the choice of $h$ is sufficient to identify discriminate the collection of distributions that we care about, it is convenient to just examine Eq. (11).

In our specific application, interaction detection, we find the embedding vector that are constructed from the final hidden layer of a classifier works reliably. We assume that this classifier is trained optimally such that, $\mathbb{P}(\delta_i | X) = \sigma(w_i^\top h(X))$, where $\sigma := \frac{\exp(x_i)}{\sum_j \exp(x_j)}$ is the softmax function, and that there are sufficiently diverse labels to ensure that this representation is identified up to a linear transformation; see Roeder et al. (2021) for details. Given a trained classifier, $\bar{h}_i := \mathbb{E}[h(X)|\delta_i] - \mathbb{E}[h(X)|\delta_0]$ is then the average of the embeddings associated with a particular knockout centered around the control wells. We find in our empirical experiments that the metric $\|\bar{h}_{ij} - \bar{h}_i - \bar{h}_j\|_2$ works well to identify highly interacting gene pairs.

We have not obtained theoretical arguments on justifying the use of optimal discriminator as the choice of test functions; we left this for future development.

## F EXPERIMENT DETAILS

In all the experiments, our MMD-based tests used the RBF and Matern 2.5 kernels, and we chose bandwidth using median heuristics. Unless otherwise stated, we estimated the KL score by first learning the three log-density ratios $\log \frac{p(x|\delta_{ij})}{p(x|\delta_0)}, \log \frac{p(x|\delta_i)}{p(x|\delta_0)}, \log \frac{p(x|\delta_j)}{p(x|\delta_0)}$ using contrastive learning (Hermans et al., 2020, NRE), and then obtaining the KL estimates via the smoothed mutual information "lower-bound" estimator (SMILE) (Song & Ermon, 2020) with clipping parameter $\tau = 5$. Detailed explanation about the NRE log-density ratio estimator and SMILE are provided in Appendix C.

For all separability tests, we trained the NRE model on all perturbation classes and then obtained the log-density ratios for each pair as introduced in Appendix C.1. For the NRE training, we evaluated multiple model architectures and optimizer step sizes, selecting the hyperparameter combination that yielded the best training accuracy. The selected model architectures and optimizer step sizes for each example are reported in the corresponding sections. We checkpointed the best model at the optimal training accuracy for further inference to obtain density ratio estimates and KL estimates. To optimize our models, we used ADAM (Kingma & Ba, 2017) with default hyperparameter settings.

All our experiments were run on NVIDIA H100 GPUs.

### F.1 SYNTHETIC EXAMPLES

For all the synthetic examples, we generated 20,000 i.i.d. samples for each perturbation class, including both single and double perturbations. The detailed data generation process for each example is provided below.

#### F.1.1 SYNTHETIC TABULAR FOR SEPARABILITY TEST

The separability scores in this example are computed using two different KL-divergence estimators: the simple KNN-based estimator and the NRE-based estimation procedure described in Appendix C. We used a 3-layer MLP with ReLU activation (hidden dimensions of 128 and 64) as the encoder for the NRE density ratio estimator. The NRE model was trained using the ADAM optimizer with a step size of 0.005 for 500 epochs and a batch size of 1024.

**Latent distribution** The latent variable $Z$ consists of 3 independent one-dimensional variables $P_1, P_2, P_3$, i.e.,

$$P_Z(z1, z_2, z_3) = P_1(z_1) \cdot P_2(z_2) \cdot P_3(z_3),$$

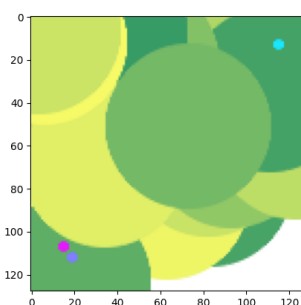 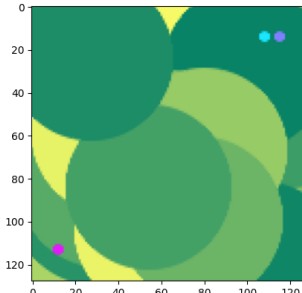

Figure 7: Example images of the interventional image data. 3 small balls with blue, purple, and red colors respectively are the targeted objects, whose locations are intervened by perturbations. Large balls with green and yellow colors form the background.

for which we consider 4 single perturbations in total, labelled as $A, B, C, D$ respectively, resulting 6 pairwise perturbations. Perturbations are applied by changing the distribution of one or more of latent variables. In our setting, perturbations are classified as separable or inseparable. Double perturbations of separable ones will intervene in two distinct latent variables, while double perturbations of inseparable ones will intervene in the same latent variable. For each node, the control (unperturbed) distribution is $\mathcal{N}(0, 1)$, while the perturbed one—whether it's single perturbed or doubly perturbed—becomes $\mathcal{N}(3, 1)$.

The association between the perturbation class and corresponding intervened latent variable(s) is described as follows:

$$
\begin{aligned}
P_1 &\sim \begin{cases} \text{Normal}(0, 1) & \text{if unperturbed} \\ \text{Normal}(3, 1) & \text{if perturbed by (A) or/and (B)} \end{cases} \\
P_2 &\sim \begin{cases} \text{Normal}(0, 1) & \text{if unperturbed} \\ \text{Normal}(3, 1) & \text{if perturbed by (B)} \end{cases} \\
P_3 &\sim \begin{cases} \text{Normal}(0, 1) & \text{if unperturbed} \\ \text{Normal}(3, 1) & \text{if perturbed by (C) or/and (D)} \end{cases}
\end{aligned}
\tag{12}
$$

As per Eq. (12), the only two inseparable pairs are A-B (both intervening $P_1$) and C-D (both intervening $P_3$).

**Sampling process** To generate observations for each perturbation class, we first generate latent samples from $P_Z$ based on the Eq. (12), and then transform the latent samples via a diffeomorphism $g(\cdot)$. In this example, we chose $g(\cdot)$ as a randomly initialized 7-layer Multi layer perceptron (MLP) with LeakyReLU activations.

### F.1.2 SYNTHETIC IMAGE EXAMPLE FOR SEPARABILITY TEST

The synthetic images consist of three objects (three small colored balls) in different locations and with various backgrounds. The positions of the objects are encoded in a 3-dimensional latent variable $Z$, which follows the identical DAG structure described in Eq. (12). Each coordinate of $Z$ corresponds to the location distribution of an object, determining the distribution of both the $x$ and $y$ coordinates of the object. Thus, a perturbation affecting the location distribution of one object will intervene in the distribution of both coordinates of that object. Background distortion is controlled by random noise. Scenes are generated using a rendering engine from PyGame, denoted as $g(\cdot)$. Example images are provided in Fig. 7.

For the NRE model, we used a 5-layer convolutional network (with channels 32, 54, 128, 256, and 512) featuring batch normalization and leaky ReLU activations as the encoder. The model was trained using the ADAM optimizer with a step size of 0.0002 for 200 epochs and a batch size of 1024.

### F.1.3 Synthetic tabular example for disjointedness test

The latent distribution is set to be a 6-component mixture distribution,

$$P_Z(z) = \frac{1}{8}P_0(z) + \frac{1}{8}P_1(z) + \frac{1}{8}P_2(z) + \frac{1}{8}P_3(z) + \frac{1}{4}P_4(z) + \frac{1}{4}P_5(z).$$

We define 7 single perturbations, labeled as A, B, C, D, E, F, G, respectively. The control distributions (unperturbed distribution) of all mixture component are $\mathcal{N}(0, 1)$. Perturbations are applied by changing the distribution of one or more of the mixture components. E.g., if perturbation $D$ is applied, $P_4 \sim \text{Normal}(0, 5)$. In our setting, perturbations are classified as independent ones and interacting ones, where double perturbation of independent ones will intervene two distinct mixture components, while double perturbation of interacting ones will intervene the same mixture component. The association of the perturbations and the corresponding component gets intervened is provided as follows:

The independent group consists of the mixture components $P_0, P_1, P_2, P_3$:

$$
\begin{aligned}
P_0 &\sim \text{Normal}(0, 1) && \text{(not intervened)} \\
P_1 &\sim \text{Normal}(5, 1) && \text{(A)} \\
P_2 &\sim \text{Normal}(10, 1) && \text{(B)} \\
P_3 &\sim \text{Normal}(-5, 5) && \text{(C)}
\end{aligned}
$$

The interacting group consists of the variables $P_4, P_5$:

$$
P_4 \sim \begin{cases}
\text{Normal}(0, 5) & \text{(D)} \\
\text{Normal}(-10, 5) & \text{(E)} \\
\text{Normal}(10, 5) & \text{(DE)}
\end{cases}
$$

$$
P_5 \sim \begin{cases}
\text{Cauchy}(15, 1) & \text{(F)} \\
\text{Cauchy}(-15, 1) & \text{(G)} \\
\text{Cauchy}(20, 1) & \text{(FG)}
\end{cases}
$$

**Sampling process**  To generate latent samples, we draw from the specified distributions and the constructed mixture model. Let $n$ be the number of samples and $d$ the dimension. The samples are generated as follows:

$$
\begin{aligned}
&\text{if no intervention:} && \mathbf{X} \sim P_0 \\
&\text{if intervention is specified:} && (\mathbf{P}_s, \mathbf{w}_s) = \text{build\_mixture(intervene)} \\
&\text{draw categorical samples:} && \mathbf{z} \sim \text{Categorical}(\mathbf{w}_s) \\
&\text{generate samples from each component:} && \mathbf{X}_i \sim \mathbf{P}_s[i]
\end{aligned}
$$

We generated latents for all 7 single perturbations, and all double perturbations, and then obtained the observations, which are used to perform tests on, by mapping the latent samples through a deterministic function $g(\cdot)$. We chose $g(\cdot)$ as a randomly initialized 10-layer MLP with LeakyReLU activations.

## F.2 Real data examples

### F.2.1 Gene-gene interactions

For the separability test, we used a 3-layer MLP with ReLU activation (hidden dimensions of 2048 and 256) as the encoder for the NRE density ratio estimator. We trained the NRE model using the ADAM optimizer with a step size of 0.0001 for 2500 epochs and a batch size of 16384.

Table 1 describes the corresponding pathways for each selected gene.

### F.2.2 Guide-guide interactions

The single-cell painting images are derived from multi-cell images, with each single-cell nucleus centered within a $32 \times 32$ pixel box. The encoder of the NRE model maps single-cell images of shape $(6, 32, 32)$ into a 128-dimensional feature vector. It consists of three convolutional blocks, each comprising a Conv2D layer with a 3x3 kernel, BatchNorm2D, ReLU activation, and MaxPool2D, progressively increasing the number of channels from 6 to 32, 64, and 128 while halving the spatial

Table 1: List of gene indices and their associated pathways

| Gene Index | Pathway |
|---|---|
| gene0 | Amino acid sensing (mTOR pathway) |
| gene1 | |
| gene2 | |
| gene3 | |
| gene4 | |
| gene5 | Apoptosis |
| gene6 | |
| gene7 | |
| gene8 | |
| gene9 | |
| gene10 | |
| gene11 | |
| gene12 | |
| gene13 | |
| gene14 | |
| gene15 | Autophagy |
| gene16 | |
| gene17 | |
| gene18 | |
| gene19 | |
| gene20 | |
| gene21 | ERAD (protein folding) |
| gene22 | |
| gene23 | Integrated Stress Response |
| gene24 | |
| gene25 | Microtubule |
| gene26 | |
| gene27 | PI3K-Akt signaling |
| gene28 | |
| gene29 | Proteasome |
| gene30 | |
| gene31 | Protein translation |
| gene32 | Protein translation (mTOR pathway) |
| gene33 | |
| gene34 | Ribosome |
| gene35 | |
| gene36 | Transcriptional regulation |
| gene37 | |
| gene38 | UPR (protein folding) |
| gene39 | |
| gene40 | |
| gene41 | |
| gene42 | |
| gene43 | |
| gene44 | mTOR signaling |
| gene45 | |
| gene46 | |
| gene47 | |
| gene48 | |
| gene49 | p53 signaling |

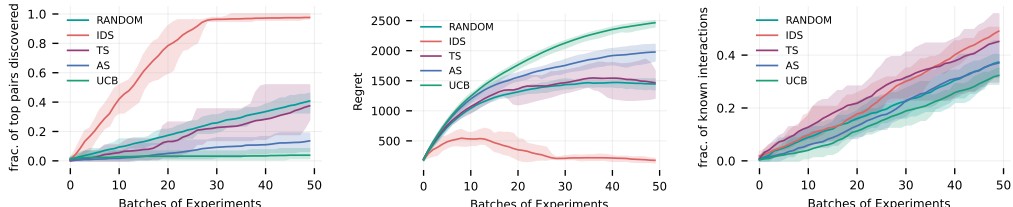

Figure 8: Empirical results on the active pairwise experiments selection for the gene-gene interaction detection example using MMD-based (using RBF kernel) test statistics. The solid lines represent the mean performance, whereas the shaded region represent all the runs (min-max). IDS (ours) outperforms all the baselines significantly in terms of top scoring pairs discovered as well as the regret. In terms of known interactions, IDS still outperforms the baselines by a small margin.

dimensions at each max-pooling step. After the convolutional layers, the output tensor of shape $(128, 4, 4)$ is flattened to $(2048)$ and passed through two fully connected blocks, each with a Linear layer, ReLU activation, and Dropout (with a dropout rate of $0.3$), transforming the feature size from 2048 to 256 and finally to 128. We train the NRE model with the ADAM optimizer, using a step size of 0.00005 for 5000 epochs and a batch size of 2048.

### F.3 ACTIVE LEARNING

**Modeling assumptions.** In terms of the specific concern about model assumptions in the active matrix completion problem for our gene-gene interaction example, we adopt the following Bayesian model to recover $\mathbf{R}$:

$$\mathbf{R} = UV^T + \epsilon, \quad U, V \in \mathbb{R}^{d \times r}, \epsilon \sim N(0, \sigma^2),$$

where $r$ is a hyperparameter smaller than $d$, encoding the low-rank structure of $\mathbb{R}$. We impose a Gaussian prior distribution on the entries of $U$ and $V$, and the inference result is characterized by the posterior distribution $p(U, V|\mathbb{R})$. The active matrix completion problem aims to approximate $p(U, V|\mathbb{R})$ without requiring all entries of $\mathbb{R}$.

As noted earlier, the low-rank assumption of $\mathbb{R}$ is justified by the morphological similarity between genes, particularly those within the same biological pathways. Our empirical results in subsection 5.2 show that gene-gene interaction scores tend to cluster by relevant pathways.

Regarding the choice of the prior, we emphasize that the validity of this Bayesian inference does not depend on $U$ and $V$ generated from a Gaussian distribution. The Gaussian prior is merely a modeling assumption for $U$ and $V$, while the posterior distribution is non-Gaussian. We remark that our ADS pipeline does not require any changes if other priors for $U$ and $V$ are adopted. Poorly specified priors might lead to suboptimal posterior characterizations of $U$ and $V$, reducing the effectiveness of the active matrix completion problem. However, our empirical results suggest that the active matrix completion method performs effectively – validating the choice of the Gaussian priors.

**Details.** To obtain samples from the posterior distribution over the low-rank reward matrix (with rank $m$), we use stochastic variational inference (Wingate & Weber, 2013; Ranganath et al., 2014), and specifically the implementation from `numpyro` (Bradbury et al., 2018; Bingham et al., 2019; Phan et al., 2019). We train the variational posterior for 5000 epochs with a learning rate of 0.01 using the Adam optimizer. We then generate $k$ samples from the posterior. For each algorithm, we run a sweep over all a set of hyperparameters. We then pick the best hyperparameters and run the experiment over 10 different seeds to get the final results. For IDS, we tune $m \in \{3, 5, 7, 10, 12\}$, $\lambda \in \{2, 3, 4, 5\}$ and $k \in \{500, 750, 1000, 1500\}$. For TS and US, we tune $m \in \{3, 5, 7, 10, 12\}$ and $k \in \{500, 750, 1000, 1500\}$. For UCB we tune $m \in \{3, 5, 7, 10, 12\}$, $\beta \in \{0.01, 0.1, 0.2, 0.5, 1, 2, 5\}$ and $k \in \{500, 750, 1000, 1500\}$, where $\beta$ controls the exploration.

## G  ADDITIONAL FIGURES

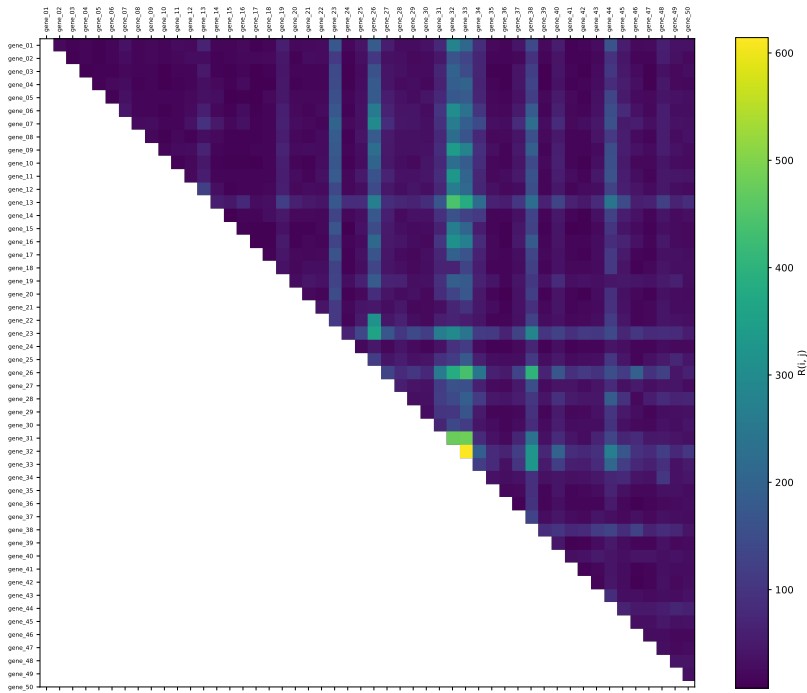

Figure 9: $l_2$ norm difference between the predicted and actual pairwise representation, $\|\vec{h}_{i,j} - \vec{h}_i - \vec{h}_j\|_2$, where the representation is defined by the final hidden layer of a classifier

