# OpenReview forum: "Automated Discovery of Pairwise Interactions from Unstructured Data"
_ICLR.cc/2025/Conference — Submitted to ICLR 2025_

### Official Review · Reviewer_hnXh · 2024-10-25

**Soundness:** 3
**Presentation:** 3
**Contribution:** 3
**Rating:** 5
**Confidence:** 3

**Summary:**

The authors propose two test statistics to measure the separability and disjointness between pairs of perturbations. Perturbations are separable if they affect disjoint subsets of latent variables, and are disjoint if they affect disjoint parts of the observation. Their separability metric is validated on two synthetic datasets where the ground-truth data generating process is known (i.e. the graph of the perturbation and latent variables), as well as on a real cell painting dataset. They also show that the disjointness metric can be used in an active learning context on cell painting data to better uncover perturbations with pairwise interactions relative to several baselines.

**Strengths:**

* The authors tackle the important problem of determining whether it is worthwhile to run a costly experiment where two perturbations are applied simultaneously.
* The evaluation is thorough, and demonstrates utility on a real-world problem.

**Weaknesses:**

* This is an applied paper, and I believe the novelty is weak.
* The separability metric essentially measures the mutual information between the latent variables corresponding to a pair of perturbations. There are many different ways to instantiate this, and it's unclear whether the authors' proposed metric itself is a valuable contribution.
* A similar statement also holds for the disjointness metric.
* Essentially, this paper combines many existing methods to achieve an empirical (and somewhat narrow) goal. I believe this is a valuable contribution, but it may be suited for a more applied venue.

**Questions:**

* Can you counter my points (in weaknesses) regarding the lack of novelty in this work?

Minor points:
* The right-hand side of the bottom equation on p.4 seems off; isn't $p_{Z_i}$ a distribution over a scalar random variable, and the argument $g^{-1}(x)$ is a vector?
* I think the clarity can be improved (and some space saved) if you apply a log to both sides of Equation (1) and go directly to Equation (3), without writing Equation (2).

---

> ### Author Response · Authors · 2024-11-21
>
> Thank you for your comments on our work. Please find our responses below.
>
> > This is an applied paper, and I believe the novelty is weak.
>
> This paper provides the first rigous definition of an interaction between treatment variables on unstructured data & it supports these definitions with two theorems. Further it provides the first active learning approach to image-based outcomes (images as inputs are well-studied) and supports this with rigous experimentation on real world data (that was specifically collected for this study).
>
> Regarding the novelty of our approach, please see our detailed point-by-point response below.
>
> > The separability metric essentially measures the mutual information between the latent variables corresponding to a pair of perturbations. There are many different ways to instantiate this, and it's unclear whether the authors' proposed metric itself is a valuable contribution. A similar statement also holds for the disjointness metric.
>
> We respectfully disagree with this point. Section 3 of our work, which introduces the separability and disjointness tests, has two key objectives for unstructured measurements: (1) to rigorously define the notion of interaction, and (2) to design measurable metrics that quantify the intensity of such interactions. To our knowledge, no prior work has addressed these objectives (though we welcome references to relevant literature if we have overlooked any).
>
> We do not claim that performing independence testing via mutual information or additivity testing via learned representations is the key novelty of our work. Instead, our primary contribution lies in rigorously characterizing the types of interactions identifiable by each method and specifying the precise modeling assumptions required for these tests. We believe the theoretical foundation provided by our work offers valuable insights for the community.
>
>
> > Essentially, this paper combines many existing methods to achieve an empirical (and somewhat narrow) goal. I believe this is a valuable contribution, but it may be suited for a more applied venue.
>
> Thank you for recognizing the value of our contribution!
>
> We want to emphasize that this work establishes a general framework for efficiently detecting (potentially rare) pairwise interactions without requiring structured measurements. Given this flexible setting, we believe our methods are broadly applicable across many scientific domains (as outlined in the first paragraph of the introduction). We do not consider this to be a narrow empirical goal.
>
> > Can you counter my points (in weaknesses) regarding the lack of novelty in this work?
>
> Please see our previous responses.
>
> > The right-hand side of the bottom equation on p.4 seems off; isn't a distribution over a scalar random variable, and the argument is a vector?
>
> If you are referring to the equation in lines 205–207, we believe it is correct. All relevant probability density functions pertain to random variables defined on general measurable spaces $\mathcal{X}$ and $\mathcal{Z}$, which are not required to be scalars.
>
> > I think the clarity can be improved (and some space saved) if you apply a log to both sides of Equation (1) and go directly to Equation (3), without writing Equation (2).
>
> Thank you for the suggestion. We will edit the relavent text in the camera-ready to improve clarity.
>
> Thank you again for your comments. We are happy to answer any further questions you have during the discussion period.

---

> > ### Comment · Reviewer_hnXh · 2024-11-25
> >
> > Thank you for your response.
> >
> > > We respectfully disagree with this point. Section 3 of our work, which introduces the separability and disjointness tests, has two key objectives for unstructured measurements: (1) to rigorously define the notion of interaction, and (2) to design measurable metrics that quantify the intensity of such interactions. To our knowledge, no prior work has addressed these objectives (though we welcome references to relevant literature if we have overlooked any).
> >
> > Thanks for the clarification, I see now that separability is different from mutual information. Seeing that it's the absolute difference between the joint and individual KL terms, there are many ways of computing it besides the approach that you tried (e.g. iVAEs). Since you're presenting a new framework, I think it's important to see the robustness of your conclusions across several implementation methods.
> >
> > I also agree with Reviewer 1aqQ, in that it's important to see your particular definition of interaction compared with some kind of baseline. While I'm not very familiar with this area, it seems implausible that this is the first time anyone has considered the notion of interaction between two vector-valued random variables.
> >
> > > If you are referring to the equation in lines 205–207, we believe it is correct.
> >
> > On the right-hand side, I believe $p_{Z_i}$ is a density over a scalar-valued random variable, and its argument $g^{-1}(x)$ is a vector-valued random variable with dimension $L$.

---

> ### Author Response · Authors · 2024-11-26
> **Thank you for your follow-up comments! Please see our response below.**
>
> > I see now that separability is different from mutual information. Seeing that it's the absolute difference between the joint and individual KL terms, there are many ways of computing it besides the approach that you tried (e.g. iVAEs). Since you're presenting a new framework, I think it's important to see the robustness of your conclusions across several implementation methods.
>
> The separability test relies on sample-based KL divergence estimates between the perturbed group and the control group. If the KL estimator is unreliable, the conclusions drawn from the separability test may also lack reliability. However, this does not reflect a lack of robustness in the testing framework itself. The key contribution of this work lies in providing a rigorous framework for understanding what to test, rather than proposing specific KL estimators---a challenging problem in its own right. Our framework is flexible and can accommodate various estimators, allowing users to select an appropriate KL estimation method based on their data type. For example, in our synthetic experiments, a simple KNN-based KL estimator proved reliable for low-dimensional observations.
>
> In our experiments, we used a KL estimation pipeline informed by recent advances in sample-based KL estimation. A detailed explanation of our approach, including the rationale for our chosen pipeline and a survey of alternative strategies from the literature, is provided in Appendix B.
>
> Additional clarifications:
>
> - The reviewer suggested using iVAE (we assume this refers to identifiable VAE; [Khemakhem et al., 2020]) in our framework. However, iVAE does not provide valid KL estimates for samples from two distributions, and we are unsure of its applicability to our approach.
> - If the reviewer is proposing latent variable identification (essentially learning causal representations for disentanglement) instead of directly testing the latent dependence from observations, we have discussed in detail in the related work section why this approach is less suitable for our purposes. It's also worth noting that iVAE requires conditionally independent latents (conditional on an environment variable) a prior, while we are tesing whether there is a dependence in the latent space.
>
> > I also agree with Reviewer 1aqQ, in that it's important to see your particular definition of interaction compared with some kind of baseline. While I'm not very familiar with this area, it seems implausible that this is the first time anyone has considered the notion of interaction between two vector-valued random variables.
>
> Please see our response to the follow-up comments from Reviewer 1aqQ.
>
> > On the right-hand side, I believe $p_{Z_i}$ is a density over a scalar-valued random variable, and its argument $g^{-1}(x)$ is a vector-valued random variable with dimension $L$.
>
> Thank you for pointing this out! This is admittedly an inprecise expression---$p_{Z_i}(g^{-1}(x))$ should be interpreted as $p_{Z_i}([g^{-1}(x)])$ where $ \[ \cdot \]\_{i} $ is the projection operator to the subspace of $Z_i$. The precise expression (updated in the manuscript) is as follows:
> $$ \frac{p(x | \delta_i)}{p(x|\delta_0)} = \frac{p_Z(g^{-1}(x)| \delta_i)\left | \text{det}(J(g^{-1}(x))) \right|}{p_Z(g^{-1}(x)|\delta_0)\left | \text{det}(J(g^{-1}(x))) \right|} =  \frac{p^\dagger_{Z_i}([g^{-1}(x)]\_i)}{p_{Z_i}([g^{-1}(x)]\_{i})}.$$

---

> > ### Comment · Reviewer_hnXh · 2024-12-01
> >
> > Thanks for your reply, and for posting an updated pdf.
> >
> > > Our framework is flexible and can accommodate various estimators, allowing users to select an appropriate KL estimation method based on their data type.
> >
> > This is the point I was trying to make - since the framework itself is the novelty of this work, I think we need to see evidence of this statement. I.e. as a potential user of this framework, it would be helpful to know whether the results will change dramatically if you swap the SMILE estimator with $\tau = 5$ with something else.

---

### Official Review · Reviewer_B1Jx · 2024-10-30

**Soundness:** 3
**Presentation:** 3
**Contribution:** 3
**Rating:** 6
**Confidence:** 2

**Summary:**

This paper proposes a method for detecting pairwise interactions between gene perturbations.
It first proposes two statistical tests for deciding whether two perturbations are separable or disjoint.
The statistical test of disjointness is employed in a greedy active matrix completion algorithm that decides which pair of perturbations to examine in the next step. The proposed method is evaluated in sythetical settings showing the validity of the statistical tests as well as in a real gene perturbation experiment where it performs better against other baselines that consider different choice of policy in the active matrix completion procedure.

**Strengths:**

The proposed methodology is interesting.
The main strength of this work is that it makes the biological experimental process more efficient and with lower costs.
The theoretical claims are justified with mathematical proofs and the effectiveness of the algorithm is empirically validated.

**Weaknesses:**

I am not a specialist in the biological field of gene perturbations experiments, but based on my understanding I would point out the following potential weaknesses for the improvement of the paper.

1. It is not quantified how much the biological experiments benefit from the active learning algorithm in terms of the total number of necessary perturbations.
How much more efficient is your algorithm compared to a standard exhaustive approach that would consider all possible perturbations?

2. I am not sure if it possible but it would make sense to additionally compare the proposed method against causal discovery methods that utilize interventions, for example, DCDI [1] or GIES [2], or methods specifically designed for gene regulatory networks [3]. It would be interesting to see what would be the performance of a causal discovery method for a subset of given perturbations. Ofcourse that would require transforming the data into a format that contains a single value for every node for each sample.

[1] Brouillard, Philippe, et al. "Differentiable causal discovery from interventional data." Advances in Neural Information Processing Systems 33 (2020): 21865-21877.

[2] Hauser, Alain, and Peter Bühlmann. "Characterization and greedy learning of interventional Markov equivalence classes of directed acyclic graphs." The Journal of Machine Learning Research 13.1 (2012): 2409-2464.

[3] Aibar, Sara, et al. "SCENIC: single-cell regulatory network inference and clustering." Nature methods 14.11 (2017): 1083-1086.

**Questions:**

I have the following questions.

1. By pairwise interactions do you mean causal dependencies or correlations? Can you also discover causal relations?
2. What do you mean by "unstructured data"? I would say that an image is structured as a grid of pixels.
3. Line 094 please elaborate on what you mean "by this notion of interaction".
4. Line 374 what are 3 dimensional tabular data? Please provide the dimensions.
5. Line 478 "Genes from the same pathway are ordered adjacently". Please rephrase/explain what it means.
6. Line 509. Can you explain more about how the top 5% pairs are chosen? Are they among the known interactions?
7. Line 511. How is the regret computed?

Some recommendations:
1. The first two sentences in the introduction require citation.
2. The font size in Fig. 1 is very small.
3. Increase the font size of the colormap in Fig. 2.
4. The font size in Fig. 3 (labels on images on the right) is very small.
5. In Fig. 3 you need to explain what the image on the right shows (presence or absence of interaction).
6. In Fig. 4 you should add a label in each figure and increase the font size of the colormap.
7. For Fig. 5 the same, nothing is visible.

I would like to hear your opinion on my feedback first and then be willing to raise my score.

---

> ### Author Response · Authors · 2024-11-21
>
> Thank you for your comments on our work. Please find our responses below.
>
> > It is not quantified how much the biological experiments benefit from the active learning algorithm in terms of the total number of necessary perturbations ...
>
> As illustrated in Figure 6, within 50 batches (i.e. 500 pairs of the possible 1225) we are able to discover 12-15% more *known* interactions than random selection. If one can run all the possible pairs, you would discover all the interaction effects. However, our focus is on efficiently discovering these interactions, which we demonstrate -- 46% interactions discovered in 40% of the total interactions.
>
> > I am not sure if it possible but it would make sense to additionally compare the proposed method against causal discovery methods ...
>
> As mentioned in the comment, causal discovery methods are not typically suitable for our problem because it assumes all variables that causally discribes the response are observed, which is not viable in our setting. Our goal is to develop techniques that apply to *unstructured data* so we do not allow the possibility of "transforming the data into a format that contains a single value for every node".
>
> Causal representation learning is more aligned with our setting. However, as we discussed comprehensively in the related work section, these CRL works all attempt to disentangle unstructured observations into latent variables, and would solve our problem if successful (one could directly observe dependencies among the inferred latents). But they depend on assumptions which are at least as strong as the assumptions that we make here, and if either the assumptions that they make or the estimation procedures they use (i.e. fitting the relevant deep nets) fail, then any conclusions will not be valid. The challenge is that it is very difficult to know whether disentanglement is successful: the assumptions are untestable, and there is no validation set metric that tells you whether you have successfully disentangled latent variables (if there was, then by definition, there would not be an identifiability problem) which makes it very difficult to reliably tune hyperparameters.
>
> By contrast, our interaction tests jointly test both the data generating process assumptions (which you don't want to test) and the independence assumptions (which you do). Our assumptions on the data generating process are comparatively mild, but even if they are incorrect, this will just result in false positives on the test rather than a failure of the inference procedure. Obviously we would prefer to avoid false positives, but in practice, a test that detects real interactions with a high false positive rate is still useful for biologists because it narrows the search space to a relatively small set of candidates.
>
> > By pairwise interactions do you mean causal dependencies or correlations? Can you also discover causal relations?
>
> As mentioned in our response to Reviewer 1aqQ, we defined two notions of interactions: violations of separability and disjointedness. While these concepts are related to causal dependencies (particularly the separability), they are not equivalent. Specifically, within the separability framework, interactions are defined as instances where two perturbations target the same latent variables. Similar concepts have been studied in the context of causal representation learning; see [1] for reference.
>
> [1] Score-based Causal Representation Learning: Linear and General Transformations
>
>
> > What do you mean by "unstructured data"? I would say that an image is structured as a grid of pixels.
>
> Unstructured data refers to measurements that do not directly reveal a specific pre-selected property. For example, if we are interested in cell viability, a structured measurement would be the number of healthy cells in a sample, whereas cell images---considered unstructured measurements---require additional processing to extract this information. For a more detailed explanation, please refer to lines 74–84.
>
>
> > Line 094 please elaborate on what you mean "by this notion of interaction".
>
> We rigorously define the two types of perturbation interactions: violations of separability (Definition 3.5) and violations of disjointedness (Definition 3.7).
>
> > Line 374 what are 3 dimensional tabular data? Please provide the dimensions.
>
> The data in this example is in $\mathbb{R}^3$. The precise data generating process is provided in Appendix E.1.
>
> > Line 478 "Genes from the same pathway are ordered adjacently". Please rephrase/explain what it means.
>
> This means that in the 2x2 matrix in Figure 5, genes from the same biological pathways are placed in adjacent rows (and columns). For the indexing of the selected genes and their corresponding pathways, please refer to Table 1 on page 24.

---

> > ### Author Response · Authors · 2024-11-21
> >
> > > Line 509. Can you explain more about how the top 5% pairs are chosen? Are they among the known interactions?
> >
> > The top 5% of pairs refer to those with the highest interaction scores, indicating a higher likelihood of interaction according to the separability or disjointedness test. This result (left subfigure of Figure 6) demonstrates that our active matrix completion method effectively identifies pairs with high interaction scores.
> >
> > However, as noted in the limitations, these high-score pairs do not necessarily correspond to known interactions.
> >
> > > Line 511. How is the regret computed?
> >
> > The regret at step i is computed as the difference between the values of the actions selected till step i and top i values in the ground truth matrix. Note that this metric is just for validating the approach and cannot be computed in practice (since the ground truth matrix is not known).
> >
> > > Some recommendations:
> >
> > We thank the reviewer for the advice! We will improve the visualization of our results in the camera-ready version.
> >
> > Thank you again for your comments. We are happy to answer any further questions you have during the discussion period.

---

### Official Review · Reviewer_1aqQ · 2024-11-03

**Soundness:** 3
**Presentation:** 4
**Contribution:** 3
**Rating:** 3
**Confidence:** 5

**Summary:**

This paper proposes a new method for exploring the structure of a black-box system of interest by introducing perturbations, specifically focusing on identifying "interacting pairs"—pairs of perturbations that yield results significantly different from the effects of each perturbation applied individually. The paper proposes two methods to examine whether given pairs of perturbations exhibit a non-trivial "interaction effect" in this sense. The first method is a "separability" test, which checks if the effects of a pair of perturbations provide information beyond that obtained from each perturbation alone. The second method is a "disjointedness" test, which quantifies whether each perturbation in the pair influences distinct subsets of the outcome space. The separability test uses a criterion based on KL divergence, while the disjointedness test uses an MMD-based two-sample test for identity. Additionally, they treat the task of calculating each test statistic for all pairs (i, j) as a matrix completion problem and apply an active learning-based sequential experimental design using ADS (Xu et al., 2022) to identify pairs (i, j) with potentially having larger test-statistics values. Experiments demonstrate the superiority of the proposed method on benchmark synthetic data that meets the study's assumptions, and they conduct a synthetic lethality test to determine if two gene knockouts result in cell lethality, empirically confirming the method’s effectiveness in real biological systems.

**Strengths:**

- The paper addresses the problem of identifying pairwise interactions, specifically highlighting cases where the effect of two perturbations, such as cell lethality from double gene knockout, is entirely different from the effects of each perturbation alone. In the experiments, gene knockout was actually performed to validate effectiveness.
- The two proposed tests are technically intriguing. Each test is well-organized with necessary assumptions and effectively leverages existing theories, incorporating reasonable methods such as MMD-based two-sample tests (Gretton, 2012) and ASD (Xu et al., 2022) to construct a effective procedure.
- Despite potentially abstract topic of identifying "interaction effects", the Introduction clearly explains the ideas and intentions. The discussion is grounded with examples, such as validation on synthetic data and actual biological applications, making the logic easy to follow.

**Weaknesses:**

- The two proposed interaction tests are not compared with any standard methods. The problem in question is not new; it has a long history in statistics as the "interaction effect," where the combination of two or more factors produces an effect greater (or less) than the sum of their individual effects [1][2]. Traditional applied statistical methods (likelihood-ratio tests, two-way ANOVA, etc) have also been used for identifying synthetic lethality [3], so a comparative analysis and discussion of differences with conventional methods are necessary. The paper only includes internal comparisons among variations of the proposed method, offering limited basis for objectively assessing its effectiveness against previous methods.
   - [1] Cox, D. R. (1984). Interaction. International Statistical Review / Revue Internationale de Statistique, 52(1), 1–24. https://doi.org/10.2307/1403235
   - [2] Amy Berrington de González, & Cox, D. R. (2007). Interpretation of Interaction: A Review. The Annals of Applied Statistics, 1(2), 371–385. http://www.jstor.org/stable/4537441
   - [3] Akimov Y, Aittokallio T. Re-defining synthetic lethality by phenotypic profiling for precision oncology. Cell Chem Biol. 2021 Mar 18;28(3):246-256. doi: 10.1016/j.chembiol.2021.01.026.

- Biologically, this issue is known as "epistasis," a well-researched topic with substantial existing literature. While statistical interaction tests have been applied, domain knowledge and modeling of underlying structures are essential (e.g., [4][5]). Although this study treats synthetic lethality as a special case and even conducted biological experiments, it does not discuss this background, leaving unclear what value this research or its new methods can provide on the top of many traditional studies on this long-standing research topic.
   - [4] Segrè, D., DeLuna, A., Church, G. et al. Modular epistasis in yeast metabolism. Nat Genet 37, 77–83 (2005). https://doi.org/10.1038/ng1489
   - [5] Terada A, Okada-Hatakeyama M, Tsuda K, Sese J. Statistical significance of combinatorial regulations. Proc Natl Acad Sci U S A. 2013 Aug 6;110(32):12996-3001. doi: 10.1073/pnas.1302233110.

- Consequently, the adaptive sampling and bandit approaches proposed in Section 4 to identify pairs with interaction effects “without directly measuring pairwise effects” are unconvincing and require additional justification, clearer assumptions, and validation. By definition, determining the presence of an "interaction effect" would seem to require measuring both pairwise and single effects at least. For example, in synthetic lethality, there are gene pairs where knocking out either gene alone is non-lethal, but a double knockout is lethal. In this case, we have no observable clues suggesting that this pair is likely to have an "interaction effect." While assumptions in Sections 3.1 and 3.2 seem related to this point, some, such as the "low-rank R with a Gaussian prior on the columns" in line 350, appear ad hoc and inadequately justified. This point requires thorough analysis and explanation for proper justification of the proposed method.

- Further, introducing bandit (specifically, best-arm identification) for this problem requires clarification. In bandit approaches, sampling each arm multiple times is generally assumed, making the theory inapplicable in settings with no sampling for many (i,j) pairs. For synthetic lethality, for instance, it would be necessary to test double knockouts across all (i, j) pairs with replicates. Figure 6 compares the method with random search and conventional bandit methods, but the meaning of this comparison is unclear, necessitating additional justification.

- While the paper aims to address general "interaction effect" identification, its validation is limited to synthetic lethality, providing insufficient evidence for its general effectiveness across other cases.

**Questions:**

- For research proposing a new method, it would be essential to compare its advantages and accuracy against existing approaches. There are various statistical methods (likelihood-ratio tests, two-way ANOVA, etc) to identify interaction effects, so did you conduct comparisons with these standard methods?

- In terms of detecting synthetic lethality, I believe there are several studies in genetic statistics related to identifying "epistasis." Did you consult this literature?

- What are the primary reasons you believe traditional methods are inadequate for this purpose and we need new methods?

- Two methods for testing the interaction effect are proposed, but their distinct purposes and value are unclear. Which method appears more promising, and under what conditions should each be used?

- How do you test for the KL divergence identity condition required for the "separability test"? Is this a simple binary decision based on whether the KL divergence is below a certain threshold, rather than a hypothesis test? If so, how did you determine the threshold?

- The theoretical justification for the entire Section 4 seems insufficient; could you provide additional support if available? The "interaction effect" as defined in this paper seems fundamentally based on comparing the effects of "double perturbation" with "individual perturbations." Given this, the rationale for identifying promising pairs with interaction effects using adaptive sampling or bandit without actually conducting double perturbations is unconvincing. If you view this as an application of bandit’s best-arm identification, typically, double perturbation across all (i, j) pairs would need to be measured several times to gain meaningful information on arm expectations. Could you provide the following information?
   - A clearer explanation of what prior information or assumptions allow us to predict interaction effects without measuring pairwise effects
   - More rigorous justification for the low-rank and Gaussian prior assumptions

- For the evaluation of the main claim of "automated discovery of pairwise interactions" as suggested in the paper's title, could you:
   - Clarify how the proposed approach handles the lack of repeated sampling for many (i,j) pairs?
   - Explain the assumptions and modifications needed to apply bandit methods in this setting?
   - Provide a more detailed interpretation of Figure 6, explaining what insights can be gained from the comparison to random search and conventional bandit methods?

- Have you conducted any validation beyond the synthetic lethality example? As the title suggests, this paper aims to propose a general method for testing "interaction effects," but with no examples beyond synthetic lethality, it might be more suitable to focus claims on identifying biological "epistasis."

- For instance, when generating test cases in software development, identifying bugs that occur only under certain condition combinations could fit within this paper’s defined "interaction effect." A system might function correctly when condition 1=True or condition 2=True, but malfunction when both are true. Do you think this method is also effective for such identification? Specifically, what is the rationale for identifying this condition combination as promising without observing the behavior under condition 1=True and condition 2=True? Would this not be challenging in black-box testing?

---

> ### Author Response · Authors · 2024-11-21
>
> Thank you for the comments on our work. Please find our responses below.
>
> > The two proposed interaction tests are not compared with any standard methods ...
>
> We thank the reviewer for these suggestions and will include a discussion in the camera-ready version. However, the first two classical methods from Cox and his coauthors do not apply in our setting because they focus scalar outcome variables whereas the **primary motivation for our paper is to ask how to generalize a notion of interaction to *unstructured* outcome data like the pixels in an image**. Our task is significantly more challenging because we not only have to contend with the high dimensionality of image data, but our notion of interaction has to be robust to arbitary bijective transformations of the data (because we cannot assume that we are in control of measurement process). To emphesize this point, consider this quote from Berrington de González and Cox [2007] which reveals just how dependent the classical approaches to interaction testing are on the judgement of a human analyst, "For a continuous and positive response variable, $y$, the transformations commonly used are logarithmic and simple powers, occasionally with a translated origin. For binary data, the logistic or sometimes probit or complementary log scale may be effective. While achieving additivity of effects is helpful, interpretability is the overriding concern. Thus, the transformation from $y$ to $y^{1/3}$ might remove an interaction but, unless y was a representation of a volume, $y^{1/3}$ might well not be a good basis for interpretation." Please refer to our response in the relevant section for further details.
>
> The third reference from Akimov [2021] serves as a fantastic motivation for the problem that we solve in this paper when they argue that, "emerging *phenotypic profiling methodologies* will improve the discovery of therapeutically relevant, novel SL interactions. [emphasis added]". They emphasize the need for more general notions of synthetic lethality (e.g. "synthetic sickness"; similar to your more general point about epistasis below), but they do not provide methods for detecting these more general notions of synthetic lethality (their Figure 4 D appears to suggest using tSNE as a way of identifying relevant outcome variables; this will lead to *very* unreliable tests!). Our interaction tests solve exactly this problem.
>
>
> > Biologically, this issue is known as "epistasis," a well-researched topic with substantial existing ....
>
> Your point that "domain knowledge and modeling of underlying structures are essential" is likely the key point of where we disagree in the value of this paper. Domain knowledge is clearly *sufficient* if it allows one to extract a well-defined outcome variable from unstructured data. But neither of the referenced papers claim it is *necessary* to detect interactions. In this paper we give two well-defined notions of interaction that apply to unstructured data. Because they do not depend on domain knowledge, they will tend to have lower specificity than tests for known interaction targets; but this also allows them to be far more scalable in detecting interactions in high-throughput screens. If every gene pair that interacted produced a different morphological phenotype, then you'd need a different test for each phenotype; whereas our tests allow you to detect interactions without prespecifying the form of the interaction.
>
> We concede that we have done a poor job of connecting with the biology literature, and we will update our references accordingly; thank you for pointing these out. That said, we emphasize that the key contribution of this work is in the interaction testing methodology itself, which we then illustrate on biological data; this is not a biology paper and does not claim to be. This is an ICLR submission: the original focus of this conference has always been on general methods for learning from unstructured data, so tests that allow us to move beyond assuming access to domain knowledge are very much the core methodological focus of the conference.

---

> > ### Author Response · Authors · 2024-11-21
> >
> > > Consequently, the adaptive sampling and bandit approaches proposed in Section 4 to....
> >
> > We believe there may be a misunderstanding here. The phrase “without directly measuring pairwise effects” doesn't appear anywhere in our paper. We can only predict pairwise effects *if* the interaction tests in Section 3 show there is no interaction, but running those tests *requires* measuring pairwise effects. However, if we are prepared to assume that the full matrix of pairwise test statistics, $R$, is low rank, then we can predict the *test statistics* of unseen pairs from what we measure in seen pairs using an active matrix completion approach. The low rank assumption simply amounts to assuming that there is some similarity in morphological effect across genes. In the absence of such structure, the procedure would indeed be no better than random. However, we believe that the low rank assumption is sufficiently justified from the empirical observations in Figure 5. The assumption on the columns being Gaussian is a variational approximation that follows prior work on IDS for this matrix completion problem.
> >
> > > Further, introducing bandit (specifically, best-arm identification) for this problem requires clarification ...
> >
> > Indeed it is true that in the traditional stochastic bandit setting, one assumes the ability to evaluate each arm multiple times. However, as we discuss in Section 4 (starting L305) we consider the discovery setting, originally proposed in [1], where each arm can be pulled exactly once and the set of actions (arms) available at each round shrinks. For this discovery setting, where each arm can only be pulled once, [1] studied the active matrix completion problem and showed that IDS achieves sublinear regret under the assumption that the matrix is low rank and the columns are drawn from a Gaussian prior.
> >
> > > For research proposing a new method, it would be essential to compare its advantages and accuracy against existing approaches ...
> >
> > We did not compare to these methods because classical statistical design of experiments approaches (typically based on factor analysis) are not applicable to our high-dimensional unstructured measurement setting. The fundamental incompatibility lies in the fact that these methods typically only work for **scalar response variables**, where relationships between the response and factors (various treatments) are modelled using linear or generalized linear regressions with explicit parametric assumptions. Interactions between factors are interpreted as non-additive effects resulting from multiple treatments, with ANOVA or likelihood ratio tests used to assess the statistical significance of such non-additivity.
> >
> > In contrast, our responses are **unstructured measurements like images**, where factor analysis fundamentally does not apply. Moreover, parametric assumptions such as normality of the response (e.g., cell images) are not sensible in this context.
> >
> > Furthermore, we want to highlight that the testing methods we employ are neither heuristic nor black-box approaches. The separability test is essentially a non-parametric likelihood-ratio test designed to evaluate the relationship described in Eq(1). Similarly, the disjointedness test is a classical two-sample test. Our key contribution lies in rigorously formulating perturbation interactions using probabilistic models, which can then be tested with well-established statistical methods.
> >
> >
> > > In terms of detecting synthetic lethality, I believe there are several studies in genetic statistics related to identifying "epistasis." Did you consult this literature?
> >
> > All of these assume domain knowledge of a scalar target variable, whereas we design tests that apply to unstructured observations. Please see our response in the weakness section.

---

> > > ### Author Response · Authors · 2024-11-21
> > >
> > > > Two methods for testing the interaction effect are proposed, but their distinct purposes and value are unclear. Which method appears more promising, and under what conditions should each be used?
> > >
> > > The two different tests target different types of interactions. Separability is somewhat more natural in that it tests whether two perturbations target the same latent variables. This is useful for discovering the target of a perturbation (e.g. a drug with an unknown target) by testing whether it interacts with a perturbation with a known target (e.g. a gene knockout).
> > >
> > > The second notion, disjointedness, has a less natural interpretation but of interest for active learning because if two perturbations are disjoint, we can predict their pairwise embedding, $h_{i,j} - h_\emptyset$, by summing the centered individual embeddings, $(h_i - h_\emptyset) + (h_j - h_\emptyset)$. While you need samples from $h_{i,j}$ to evaluate this score, by learning a posterior over the disjointness score using active matrix completion, you can predict for which $i, j$ pairs $h_{i,j}$ can be estimated using single perturbations.
> > >
> > > Both scores can be used in an active learning pipeline, but they have different use cases. If your goal is finding pairs of perturbations that interact, separability has a more natural interpretation; if your goal is predicting pairwise embeddings, disjointness is better.
> > >
> > >
> > > > How do you test for the KL divergence identity condition required for the "separability test"? ...
> > >
> > > This is a great question, and we will clarify it in the camera-ready version. We did not establish a rigorous rejection criterion for the KL-based test. While this is less critical when the goal is to use the test statistic for active learning, we agree that a formal hypothesis testing framework is important from a statistical perspective. This issue has been addressed in a follow-up work.
> > >
> > >
> > > > The theoretical justification for the entire Section 4 seems insufficient; could you provide additional support if available? ...
> > >
> > > Please see our response above.
> > >
> > > > A clearer explanation of what prior information or assumptions allow us to predict interaction effects without ...
> > >
> > > The low-rank assumption comes from the fact that there are correlations in the functional behavior of different genes. The assumption of Gaussian prior on the columns was made to follow prior work [1]. See response in weaknesses for further details.
> > >
> > > [1] Xu, Z., Shim, E., Tewari, A., & Zimmerman, P. (2022). Adaptive sampling for discovery. Advances in Neural Information Processing Systems, 35, 1114-1126.
> > >
> > >
> > > > Clarify how the proposed approach handles the lack of repeated sampling for many (i,j) pairs? Explain the assumptions and modifications needed to apply bandit methods in this setting?
> > >
> > > Please see our responses above.
> > >
> > > > Provide a more detailed interpretation of Figure 6, explaining what insights can be gained from the comparison to random search and conventional bandit methods?
> > >
> > > The goal of the experiment in Figure 6 is two fold - the first is to validate the active matrix completion approaches for the test statistic matrix and second to validate the overall approach for discovering known biological interactions. The left and right panels demonstrate that IDS is able to discover the top values of the test statistic effectively. IDS is able to find all the top 5% values of the test statistic matrix within 50 rounds (i.e. 500 experiments out of the possible 1225). The panel on the right illustrates the number of known biological interactions discovered. This validates whether the test statistic captures the biological interactions faithfully. We find that IDS which finds the top values of the test statistic indeed finds a larger fraction of the known interactions, although the margins are smaller.

---

> > > > ### Author Response · Authors · 2024-11-21
> > > >
> > > > > Have you conducted any validation beyond the synthetic lethality example? As the title suggests, this paper aims to propose a general method for testing "interaction effects," but with no examples beyond synthetic lethality, it might be more suitable to focus claims on identifying biological "epistasis."
> > > >
> > > > First, we’d like to clarify that our gene-gene interaction example is not limited to synthetic lethality. The two testing procedures aim to detect general interactions defined by violations of separability and disjointedness. Biological interactions that do not align with these definitions are not expected to be identified by our approach. It occurs that many of the gene-gene interactions identified in our experiments fall under synthetic lethality relationships, suggesting that this relationship is well captured by the two probabilistic models of pairwise interactions. The challenge of going beyond synthetic lethality is primarily in finding good sources of known biological relationships (though epistasis is a great suggestion). Pairiwise morphological interactions are not well characterized in the biology literature because without automated dection methods like the one we propose, going beyond cell viability requires a human to select & measure a morphological target of interest.
> > > >
> > > > We did however validate both testing procedures on several synthetic datasets in Section 5.1 and tested the separability of interactions between multiple CRISPR guides targeting the genes TSC2 and MTOR (see the left panel of Figure 3).
> > > >
> > > > > For instance, when generating test cases in software development, identifying bugs that occur only under certain condition combinations could fit within this paper’s defined "interaction effect." A system might function correctly when condition 1=True or condition 2=True, but malfunction when both are true. Do you think this method is also effective for such identification? Specifically, what is the rationale for identifying this condition combination as promising without observing the behavior under condition 1=True and condition 2=True? Would this not be challenging in black-box testing?
> > > >
> > > > This sounds like an interesting application!  While black-box testing for software bugs falls outside our area of expertise, if the software output measurements can be modelled using either the separability or disjointedness framework, we believe our approach could be applicable.
> > > >
> > > > Thank you again for your comments. We are happy to answer any further questions you have during the discussion period.

---

> ### Comment · Reviewer_1aqQ · 2024-11-22
>
> Thank you for your detailed response. I believe I now understand the main points of disagreement, but let me further make sure three key points as below.
>
> Rating wise, since we are reviewing the submitted manuscript, not any unseen revised versions, I keep the scores as is.
>
> ### (1) Lack of comparisons with other methods
>
> The response to this issue remains unconvincing. I do not believe it is impossible to compare your method with others.
>
> While your paper refers to “unstructured data,” it ultimately converts image features into embedded vectors using a pre-trained model—essentially transforming them into fixed-dimensional multivariate data—which are then used for interaction testing. As other reviewers may have pointed out, statistical interaction tests can be applied to multivariate data. Hence, a lack of comparisons with alternative methods is not justified.
>
> ### (2) Identifying pairs with interaction effects “without directly measuring pairwise effects”
>
> I guess that the “Automated discovery” in the title refers to the explanation in L95 of the paper: "We can search the space of pairwise experiments by selecting **pairs of perturbations that are likely to result in large test statistics**. In doing so, we reduce the problem of finding interacting pairs of perturbations into an active matrix completion problem."
>
> If I understand correctly, this means predicting test statistics for **unmeasured pairs (k, l)** based on observed perturbation results for other pairs (i, j). This would imply predicting pairwise effects for (k, l) **without directly measuring their double perturbation effects**, wouldn't it? This is the point I find the most difficult to fully accept regarding the validity of the setup.
>
> From the author response, I guess that this is possible due to assumptions employed in an existing work [1], such as the matrix being low-rank and columns following a Gaussian distribution. These assumptions are briefly mentioned in L349 of your manuscript. But [1] would not specifically target the one of this paper, i.e., pairwise interactions like synthetic lethality, simply relying on existing assumptions should require careful validation and discussion within the paper. Are these assumptions not something that can be derived from Assumptions 3.1, 3.2, and 3.4, but rather supported only by the empirical fact that the experimental results seem to work well? The paper feels theoretically solid in parts, but the critical sections come across as very loose.
>
> This gives the impression that this paper just borrows the established framework [1] for "automated discovery" without fully assessing whether its technical assumptions (e.g., low-rank matrix structure, Gaussian-distributed columns) are appropriate for this paper's research goal. From its definition, the pairwise effects this paper targets are seemingly hard to predict by partial observations of other pairs, and thus careful validation and discussion of these assumptions seem necessary.
>
> [1] Adaptive sampling for discovery. NeurIPS 2022, 35, 1114–1126.
>
> ### (3) "domain knowledge and modeling of underlying structures are essential"
>
> My comment on this point might have been confusing. I did not mean to refer to any conclusions from a specific study. Rather, I meant that prior research on detecting epistasis (or synthetic lethality as a specific case) often incorporates additional information, such as PPI, GO, pathway data, or knowledge graphs. See the following, for example.
>
> - [2] Benchmarking machine learning methods for synthetic lethality prediction in cancer. Nat Commun 15, 9058 (2024). https://doi.org/10.1038/s41467-024-52900-7
> - [3] Discovery of synthetic lethal interactions from large-scale pan-cancer perturbation screens. Nat Commun 13, 7748 (2022). https://doi.org/10.1038/s41467-022-35378-z
>
> This is because from the definition, synthetic lethality is hard to identify only from the indirect observations. Predictions therefore usually require additional assumptions or information.
>
> In this sense, the additional assumptions in this paper would be the one from [1], i.e. low-rank matrix structure, Gaussian-distributed columns. But this point should have been more carefully verified and discussed in the paper. Plus, further validation, discussion, and comparisons with existing methods seem necessary. For reference, [2] includes benchmarking of three matrix factorization-based methods (SL2MF[4], CMFW[5], GRSMF[6]).
>
> - [4] SL2MF: Predicting synthetic lethality in human cancers via logistic matrix factorization. IEEE/ACM Trans. Comput. Biol. Bioinform. 17, 748–757 (2020).
> - [5] Predicting synthetic lethal interactions using heterogeneous data sources. Bioinformatics 36, 2209–2216 (2020).
> - [6] Predicting synthetic lethal interactions in human cancers using graph-regularized self-representative matrix factorization. BMC Bioinform. 20, 1–8 (2019).

---

> > ### Author Response · Authors · 2024-11-26
> > **Thank you for your follow-up comments! Please see our response below.**
> >
> > > (1) Lack of comparisons with other methods
> > The response to this issue remains unconvincing. I do not believe it is impossible to compare your method with others. ...
> >
> > Thank you for the followup. This seems to be our key point of disagreement: one should *never* compare statistical tests with two *different* null hypotheses with respect to how the tests perform on a single dataset, because then you are jointly testing the utility of the respective null hypothesis and the statstical efficiency of a test, and your results only apply to *that* dataset.
> >
> > The problem with the procedure that you outline is that the null hypothesis of standard tests are typically defined with respect to your representation. For example, if $z = h(x)$ is your "fixed-dimensional multivariate data", then the null hypothesis is implicitly defined with respect to $h$. Our tests are **non-parametric** statements about the relationship between distributional assumptions and their associated null hypotheses (i.e. they do not depend on $h$ beyond ruling out any $h$ that throws away information). Any comparison between a test that depends on a particular choice of $h$ and one that is agnostic to $h$ (i.e. our non-parametric tests) is attempting to compare the outcome of two tests that have *two different null hypotheses*, which will be deeply misleading in general.
> >
> > While you can use data to compare the statistical efficiency of a set of tests that have the same null hypothesis, one should *never* choose a null hypothesis itself as a function of what works best on one particular dataset because that can lead to conclusions that only hold for that particular data and embedding function. This is especially misleading when one test explicitly depends on the choice of embedding function. For example if we were to discover [standard test x] on an autoencoder's embeddings leads to better discoveries of known biology, all we could conclude is that this is true for *this dataset* and *this autoencoder* (standard autoencoder embeddings are not identified, so the conclusion may even change between random seeds of the autoencoder's training). By constrast our theory is general: it gives explicit well-defined notions of depences that do not depend on your choice of representation and explains how these can be tested. The biological application serves as an instance of a class of interaction testing problems that we could (in principal) solve with this method.
> >
> > To be clear, we are not at all opposed to using existing nonparametric tests: in section 3.2 we use Maximum Mean Discrepancy (MMD) tests to test disjointness, and we are currently working on followup work that tests separability using Fisher Divergences and Kernalized Stein Discrepancies [Liu et al 2016]. All of these tests are comparible because they test the same null hypothesis (i.e. theorem 3.6).
> >
> > Regarding the statistical interaction tests mentioned by the reviewer, we are unsure which specific methods are being referred to. As previously noted, the multivariate factor analysis-based tests suggested, such as those in Berrington de González and Cox [2007], are not applicable to our setting. These methods require **scalar response variables**, where relationships between the response and factors (various treatments) are modeled using linear or generalized linear regressions. In contrast, our setting involves **multivariate response variables**, rendering those methods inapplicable. I.e. in a standard regression of the form $y = f(x_1, x_2)$ we measure a high dimension $y$ not a high-dimensional $x$. The only two recent papers that we can find that apply to high dimensional responses are "Sparse Learning and Structure Identification for Ultrahigh-Dimensional Image-on-Scalar Regression" [Li et al 2021] and "Interaction pursuit in high-dimensional multi-response regression via distance correlation" [Kong et al. 2022], neither of which apply to our setting because they both have strong linearity assumptions (but both emphasize the same point we make, which is that this area has been understudied).
> >
> > Moreover, we'd like to clarify that having a fixed-dimensional observation does not necessarily imply that it is *structured* with respect to the interventions. In many cases, the influence of interventions is entangled, inducing distribution shifts across all dimensions. A structured transformation would ideally disentangle the measurements into a set of causal latent variables, enabling direct observation of dependencies between two single perturbations among the inferred latents. In contrast, our framework does not rely on such disentanglement (see our discussion in the related work section and lines 192–193).

---

> > > ### Author Response · Authors · 2024-11-26
> > >
> > > > (2) ...
> > > > (3) ...
> > >
> > >  We agree that rigorously understanding when *prediction of test statistics for unmeasured pairs (k, l) based on observed perturbation results for other pairs (i, j)* is possible, and ensuring that the active data acquisition pipeline works effectively, are critical research directions. Validating these conditions for the gene-gene interaction problem is indeed valuable. However, each of these questions is complex enough to warrant its own dedicated research project, which is beyond the scope of this work. Our primary contribution is establishing the theoretical and methodological framework for quantifying pairwise interaction effects from unstructured observations and empirically demonstrating the feasibility of active data acquisition to optimize experimental budgets. As discussed in previous responses, these challenges are well-known in the literature but remain largely unresolved, further underscoring the value of our contribution.
> > >
> > > In terms of the specific concern about model assumptions in the active matrix completion problem for our gene-gene interaction example, we adopt the following Bayesian model to recover $\mathcal{R}$:
> > > $$\mathcal{R} = U V^T + \epsilon, \quad U, V \in \mathbb{R}^{d \times r}, \epsilon \sim N(0, \sigma^2),$$
> > > where $r$ is a hyperparameter smaller than $d$, encoding the low-rank structure of $\mathcal{R}$. We impose a Gaussian prior distribution on the entries of $U$ and $V$, and the inference result is characterized by the posterior distribution $p(U, V| \mathcal{R})$. The active matrix completion problem aims to approximate $p(U, V| \mathcal{R})$ without requiring all entries of $\mathcal{R}$.
> > >
> > > As previously explained, the low-rank assumption of $\mathcal{R}$ is justified by the morphological similarity across genes, particularly those within the same biological pathways. Our empirical results in Section 5.1 show that gene-gene interaction scores tend to cluster by relevant pathways (see lines 446–456). We have included this discussion in Line 350 --- 351 in updated manuscript.
> > >
> > > Regarding the choice of the prior, we emphasize that the validity of this Bayesian inference does not depend on $U$ and $V$ being generated from a Gaussian distribution. The Gaussian prior is merely a modeling assumption for $U$ and $V$, while the posterior distribution is non-Gaussian. We remark that our ADS pipeline does not require any changes if other priors for $U$ and $V$ are adopted. Poorly specified priors might lead to suboptimal posterior characterizations of $U$ and $V$, reducing the effectiveness of the active matrix completion problem. However, our empirical results suggest that the active matrix completion method performs effectively.
> > >
> > > We have now added this clarification in Appendix E.3.

---

### Official Review · Reviewer_GVrM · 2024-11-04

**Soundness:** 3
**Presentation:** 3
**Contribution:** 3
**Rating:** 5
**Confidence:** 3

**Summary:**

This paper presents a novel active learning method to discover pairs of highly correlated variables. Concretely, we imagine that an experimentalist has access to an observations space $\mathcal{X}$, which change as a function of perturbations. Crucially, some pairs of perturbations $\delta_i, \delta_j$ jointly operate ($\delta_{ij}$) differently on $\mathcal{X}$ relative to their marginal individual effects. The key idea of this paper is to estimate three densities — $p(x|\delta_i), p(x|\delta_j), p(x|\delta_{ij})$ — which capture the distribution of $x$ based on marginal and joint perturbations and compare them to $p(x|\delta_0)$, the world in which no perturbations are applied.

The authors formulate their active matrix completion problem in a statistical hypothesis testing framework. In particular, at each step, they estimate a posterior over $\mathbf{R}$, a matrix of test statistics for all pairs of perturbations, and greedily selects a batch of experiments to run in the next step.

The authors apply this framework to CRISPR knockouts on pairs of 50 genes. Their dataset includes observations for all pairs in this 50 gene set, so the authors simulate an active experimental design by assuming the gene pair perturbations observations are masked and then simulating matrix completion by progressively revealing observations. Their method, which they term IDS, quickly discovers top gene pairs relative to standard baselines, and minimizes natural errors like regret.

**Strengths:**

This paper is well-written and proposes an interesting method for active experimental design. Estimating the joint density and comparing to the marginal density is an interesting idea, both for the disjointedness analysis and for the separability analysis.

Although prior work (e.g., GEARS) has formulated CRISPR pair perturbation response prediction as a matrix completion problem, the active learning formulation seems novel, especially in dealing with imaging modalities (as opposed to RNAseq). The authors additionally spell out their assumptions — such as a set of latent variables $Z$ which capture all the information about perturbations — clearly. Their statistical testing framework is clear and well-motivated.

The authors additionally compare to plenty of baselines on their experiments. Although they only perform experiments on one real-world dataset, experimentation in this domain is expensive, so this is a natural choice and should not be seen as a limitation of their work.

 The appendix thoroughly explains the architecture and the methodological choices.

**Weaknesses:**

The use of a statistical test for matrix completion is a little poorly motivated. In particular, given an n x n matrix completion problem, at least one pair is likely to have a large test statistic. Given this is an active learning problem, multiple hypothesis testing is not as core of a concern, but I'd like the authors to discuss in their rebuttal what a "null" set of interaction pairs might look like. Concretely, suppose that in a module of 50 genes (so 1225 gene pairs) only 5 gene pairs interact with each other. Can this method be used to decide a "stopping" criterion for experimentation — i.e. would you be able to note that the posterior on the test statistics obey the null distribution once those 5 gene pairs have been found. Could you:
1. Describe the null distribution of test statistics in this framework
2. Discuss how this method could be used to decide a stopping criterion


Second, I think the authors did not explain why one would use imaging data for this matrix completion problem, instead of the more structured RNAseq data. Could you please justify why RNAseq data isn't used instead of imaging data?

Third, I did not get a sense of how often the joint distribution differs from the product of the marginals in this setting. There is some prior work that very few perturbations actually obey this relation, e.g. https://www.biorxiv.org/content/10.1101/2024.09.16.613342v1.full.pdf. How often is this method necessary, and why can't we just take experiment on the pairs $\delta_ij$ for which $\delta_i, $\delta_j$ are marginally most different from $\delta_0$. Could you provide quantitative analysis on how often joint distributions significantly differ from products of marginals in this dataset?

**Questions:**

Q1) See weaknesses section on detecting that all relevant gene pairs have been found

Q2) How many gene pairs usually have non-linear interactions? Or interactions that are significantly different from the raw pair?

Q3) Please include a baseline in which we reveal $\delta_ij$ on the pair $i,j$ for which $\delta_i, $\delta_j$ are marginally most different from $\delta_0$, perhaps taking the sum of the log density ratios.

Q4) How many samples are required for Appendix B.1 to become a faithful estimator of the densities? What happens if our densities are very far off?

---

> ### Author Response · Authors · 2024-11-21
>
> We thank you for the helpful feedback on our work. Please find our responses below.
>
> > The use of a statistical test for matrix completion is a little poorly motivated. In particular, given an n x n matrix completion problem, at least one pair is likely to have a large test statistic ...
>
> This is a great question and worth discussing in more detail. Our original motivation for working on this problem was thinking about exploring, e.g. all pairwise knockouts of the human genome (approx. 20 000 genes, so at least 200 million experiments). In spaces this large, you're always bounded by budgets and so optimal stopping is not a concern.
> That said, we have followup work to this which considers the problem of how to make single perturbation experimentation more efficient [no citation; currently under review]. In both this paper and that work, our acquisition function essentially optimizes for examples where we are likely to make incorrect predictions (under the disjointness statistic; see section 3.2). Because of this, if our posterior over $\mathbf{R}$ is calibrated (this can be relaxed to weaker notions of calibration), then for each batch of samples, $B$, that you select at each round, the average loss over the batch, $\frac{1}{|B|}\sum_{i,j \in B}\|\vec{h}_{i,j} - (\vec{h}_i +\vec{h}_j)\|$ is a upper bound on the average loss of IID samples from $\mathbf{R}$, and hence can be used to construct a explicit stopping criterion. Intuitively, this can be thought of as, "if you're accurately predicting all the pairs you think are *most likely to interact*, then there probably are no interactions left in the matrix." For this approach to work, you need large enough batch sizes, but in practice batch sizes tend to be large to keep experimental costs per sample low.
>
>
> > Second, I think the authors did not explain why one would use imaging data for this matrix completion problem, instead of the more structured RNAseq data.
>
> As explained in lines 74–84, unstructured measurements, such as cell microscopy images, can be acquired through automated high-throughput perturbation platforms, making them a significantly cheaper alternative to structured measurements like RNA-seq. While cell images may provide less precise signals for a specific pre-selected properties, they retain rich biological information [see Celik et al. 2024 for a comparision showing the similarity between RNA-seq and microscopy data]. The challenge, of course, is that we need methods that are able to work with unstructured data directly (rather than rely on preprocessing or specially designed assays). To our knowledge, this is the first work the provides a rigorous approach to achieving this goal of detecting interactions from unstructured data. **As such, we view the interaction tests as our primary contribution.** We performed inference on these unstructured measurements (microscopy images) to demonstrate that these test work even in the extremely challenging real world setting of microscopy imaging.
>
> > Third, I did not get a sense of how often the joint distribution differs from the product of the marginals in this setting...
>
> Thank you for bringing up this very relevant work. Ahlmann-Eltze et al [2024] is concurrent work (note that it was published the week before the ICLR abstract deadline), but they arrive at a very similar conclusion to our original experiments that motivated our disjointness test. The additive model that outperforms the deep learning based approaches in Ahlmann-Eltze et al [2024] is, $x_{i,j} = x_i + x_j − x_{0}$ (see equation 2 in [Ahlmann-Eltze 2024]). Because they are working with bulk expression data, this should be interpreted as $E[x|\delta_{i,j}] = E[x|\delta_i] + E[x|\delta_j] − E[x|\delta_{0}]$ in our notation (i.e. $x_i$ corresponds to the average expression from knockout $i$). This is just a special case of equation 5 in our paper, when you choose the embedding function $h(\cdot)$ as the identity function. Thus, our disjointness test can be interpreted as a test for when the additive model in Ahlmann-Eltze [2024] will provide good predictions of double knockouts. It is also much more general, because it also provides sufficient conditions for additivity for *any* embedding function, $h(\cdot)$. We use this observation as a starting point to our method: rather than try to predict double perturbation outcomes (which as Ahlmann-Eltze [2024] point out, has been fairly unsuccessful thus far), we aim to experiment only where this assumption fails. The heat maps in Figure 5 left and middle show two different estimates of where it fails. Consistent with Ahlmann-Eltze [2024], there is a lot of "dark blue" in the matrix where the additive model would work well, but there are also many places where it fails.

---

> > ### Author Response · Authors · 2024-11-21
> >
> > > Q1) See weaknesses section on detecting that all relevant gene pairs have been found
> >
> > Please see our response above.
> >
> > > Q2) How many gene pairs usually have non-linear interactions? Or interactions that are significantly different from the raw pair?
> >
> > Please see our response above.
> >
> > > Q3) Please include a baseline in which we reveal $\delta_{ij}$ on the pair $i,j$ for which $\delta_i,\delta_j$ are are marginally most different from $\delta_0$, perhaps taking the sum of the log density ratios.
> >
> > As clarified above, we focus on detecting when this double perturbation result **cannot** be recovered from adding single perturbations. Hence we think this baseline is not relavent to our objective.
> >
> > > Q4) How many samples are required for Appendix B.1 to become a faithful estimator of the densities? What happens if our densities are very far off?
> >
> > We agree that sample complexity is an important concern here. Our methods do not require learning the full densities; instead, we estimate the KL divergence by learning density ratios, which is generally easier because you are essentially learning a classifier. In all our examples, we train the density ratios using a few thousand images for each perturbation class, which is reasonable in cell imaging examples.
> >
> > We think that a theoretical treatment of the sample complexity of these methods is an important future direction, but we emphasize that the KL estimator itself is not our primary contribution. Our main contribution lies in showing *why* the KL divergence can be effectively used for interaction detection. More efficient of this divergence will naturally lead to better tests.
> >
> > Thank you again for the comments on our work. We are happy to answer any further questions you have during the discussion period.

---

> > > ### Comment · Reviewer_GVrM · 2024-11-22
> > > **Thank you for the response, a few follow-ups**
> > >
> > > Thank you for your response. I have a few follow up questions:
> > > 1. I am slightly confused about the notion "if you're accurately predicting all the pairs you think are most likely to interact, then there probably are no interactions left in the matrix." You claim that if you are accurately predicting the loss $\frac{1}{|B|}\sum_{i,j \in B}|\vec{h}_{i,j} - (\vec{h}_i +\vec{h}_j)|$, you have likely found most of the pairwise interactions. My confusion is regarding this claim. Suppose during round 0 (i.e., before any data acquisition) your predictor/posterior is perfectly calibrated. In that case, we'd accurately predict the loss for most pairs, so would we stop before any data acquisition? On the other hand, if the posterior is very poorly calibrated, your false discovery rate would be very high. I am unsure how to think of the tradeoff between these things. Could you clarify?
> > >
> > > 2. Thank you for the pointer to Celik et al 2024 on microscopy data! It's interesting to see this applied to gene perturbations.
> > >
> > > 3. Thank you for engaging with my concern on how often the product of the marginals differs from the joint. I agree that Ahlmann-Eltze et al [2024] is concurrent and thus this paper cannot be judged against that one. However, my broader issue is how necessary it is to model non-linear interactions of perturbation effects. Specifically, I'd like to see a quantitative estimate (or an argument for why such an estimate can't be derived) for how often the model diverges from the linear model. This is an important question, even if the linear model is a special case of your model, as if, say, 99% of interactions are linear, then a linear model could be a strong "inductive bias" that helps more efficiently discover pairwise interactions. The authors state that "we focus on detecting when this double perturbation result cannot be recovered from adding single perturbations," but I still don't understand how often this occurs. If this can be clarified, and if the reviewers can run this baseline of reveal ing$\delta_{ij}$ on the pair $i,j$ for which $\delta_i,\delta_j$ are are marginally most different from $\delta_0$, then I'd be willing to raise my score.

---

> > > > ### Author Response · Authors · 2024-11-26
> > > > **Thank you for your follow-up questions! Please see our response below.**
> > > >
> > > > > 1. ...
> > > >
> > > > Thank you for following up; we should have been more precise.  It is worth clarifying that there are two models:
> > > > 1. [Additive] The simple additive model $\hat{h}\_{i,j}: R^d \times R^d \rightarrow R^d$ which is defined as $\hat{h}\_{i,j}:= \vec{h}\_i + \vec{h}\_j$ where $\vec{h}\_i := E[h(x)|\delta_i]-E[h(x)|\delta_0]$. This simple "model" has no trainable parameters (making it trainable is an obvious extension, but in the settings that we consider $d = 1024$ and we only see 1225 unique $i,j$ pairs, so overfitting is a real concern).
> > > >
> > > > 2. [Proxy] The matrix completion model $P(R|H\_t = \\{a^{(k)}, \mathbf{R}\_{i^{(k)}, j^{(k)}}\\}\_{k=1}^{t-1}$) that aims to predict $R_{i,j} = \|\|h\_{i,j} - \hat{h}_{i,j}\|\|$.
> > > >
> > > > [Proxy] is the model that we referred to as being calibrated. Calibration is typically presented from a frequentist perspective as, $P(Y = \hat{Y}| \hat{P} = p) = p\quad \forall p\in[0,1]$ (see e.g. Guo [2017], "On the Calibration of Modern Neural Networks"), where $Y$ corresponds to $R$ in our setting (to avoid needing to be careful about densities, assume to this discussion that $P(R|H_t)$ outputs a discrete distribution over a sufficiently fine gained binning of rewards even though they are really continuous). Note that this doesn't require perfect predictions, only that the model correctly "knows" where it is uncertain (as mentioned in our previous response, this can also be weakened to a weaker notion of calibaration, but that requires a theorem). With this in mind, a model that has a uniform distribution over all possible rewards at time 0 is still calibrated (because it "knows" nothing other than its priors). The calibration assumption simply rules out the possibility that the proxy model is confidently wrong about the additive model for any $i,j$ pair.
> > > >
> > > > Now, in the special case where $R_{i,j}$ is perfect for all remaining $i,j$ pairs, one can directly measure this at every round by observing zero loss from $\frac{1}{|B|}\sum\_{i,j \in B}\|\|\vec{h}\_{i,j} - (\vec{h}\_i +\vec{h}\_j)\|\|$ at every round of experimentation. The fact that IDS is optimizing for the worse case examples given the proxy's predictions while taking into account (calibrated) uncertainty, means that getting zero loss is a reliable upper bound on the real error $\frac{1}{|\Delta_t|}\sum_{i,j \in \Delta_t}\|\|\vec{h}_{i,j} - (\vec{h}_i +\vec{h}_j)\|\|$ (i.e. the error in all remaining pairs).
> > > >
> > > > To be clear - this argument is from followup work so we do not plan to include it in this paper (it obviously needs more detail to be made formal), but we inlcude it here to make it clear that it is possible to derive a stopping criterion if needed.
> > > >
> > > > > 3. ....
> > > >
> > > > How accurately the double perturbation can be predicted with the additive model will vary as a function of your encoder, $h(\cdot)$. Trivially, if $h(\cdot) = 0$ (i.e. an encoder that maps all images to the zero vector) will be perfectly accurate for all pairs $i,j$ (because $h_{i,j} = h_i + h_j = 0$ for all $i,j$ with the trivial encoder), but useless in practice. As a result, the best way to answer your question, is to examing figure 5 (left) and (middle) which test the sufficient condition for *any* encoder to be additive (i.e. consider the $h$ that makes the error of the additive model largest). In those plots, dark blue regions correspond to areas where the additive model is accurate and green / yellow regions are where it is incorrect.
> > > >
> > > > If you would prefer to see this for a particular choice of encoder, we have now also included a plot of the $\|\|\vec{h}\_{i,j} - \vec{h}\_{i} - \vec{h}\_{j}\|\|_2$ in the appendix (final page). You will see that the plot is qualitatively similar to the MMD-based plots in figure 5. This is as expected for a representation that does not uncessarily throw information away. The accuracy of the additive model in this setting will be a function of your chosen threshold below-which you decide a prediction is "accurate". We will include ROC curves in the final version of this, but it is clear from these plots that the additive model is certainly *not* accurate everywhere. This is unsurprising from a biological perspective: it is well known that certain effects only show up in double perturbations: see for example, Reviewer 1aqQ's discussion on the extensive biological literature on 'epistasis' which studies these phenomena.
> > > >
> > > > Finally, we emphasize that the contribution of our work remains significant even if such non-linear pairwise interactions occur very rarely. It is precisely the rarity of these interactions that underscores the importance of an efficient active acquisition process to detect them without an exhaustive search over all pairs. And, if these interactions are in fact rare, then it is plausible that with methods like those we propose, we could discover all of them using active learning (if everything interacted, this would require far more extensive experimentation).

---

> ### Comment · Reviewer_GVrM · 2024-11-26
> **Confusion regarding plot**
>
> [EDIT: just saw the note to all reviewers about working on a new pdf — no rush, thank you!]
>
> Thank you for your extensive response. I am still slightly confused about what you mean by a "calibrated" posterior over the matrix $\mathbf{R}$. The typical notion of calibration — which you gave — is a measure of how far P(Y=y|f(x) = p) typically is from p. As you note, this is measured empirically in classification settings. However, in your setting $\mathbf{R}$ is a continuous density, so I am unsure what you mean by a "calibrated" posterior in the language of, e.g., Guo 2017. You seem to hint at it with your comment on "outputs a discrete distribution over a sufficiently fine gained binning of rewards even though they are really continuous." Could you please clarify this comment? I understand the general intuition, however, that calibration is a weak condition and that a calibrated model could be inaccurate on individual instances, so calibration doesn't necessarily imply stopping at step 0.
>
> The second point is interesting. However, I am unable to see any appendix figure in the PDF mirroring Fig. 5. I believe the PDFs can be updated until tomorrow — if a PDF with a fresh copy of this Figure is uploaded soon and convincingly makes the case that the additive model is insufficient in large regions of the matrix, I would be willing to raise my score.

---

> > ### Author Response · Authors · 2024-11-26
> >
> > Thanks for following up so quickly!
> >
> > > However, in your setting is a continuous density, so I am unsure what you mean by a "calibrated" posterior in the language of, e.g., Guo 2017. You seem to hint at it with your comment on "outputs a discrete distribution over a sufficiently fine gained binning of rewards even though they are really continuous." Could you please clarify this comment? I understand the general intuition, however, that calibration is a weak condition and that a calibrated model could be inaccurate on individual instances, so calibration doesn't necessarily imply stopping at step 0.
> >
> > The most direct analog for continuous random variables would be to check calibration of the cumulative distribution function, i.e.  $P(Y < x| \hat{F}(x) = p) = p\quad \forall p\in[0,1], x\in R$, but that is a far stronger condition than we need. In practice, if one cares about stopping, it would make sense to discretize $\mathbf{R}$ with a threshold, $\tau$, below which you deem the additive model to be "accurate". I.e. redefine $\mathbf{R}$ as a binary matrix with $R_{i, j} = \mathbf{1}( |h_{i,j}-h_i-h_j| < \tau)$ [those should be \vec{h} but the \vec command is causing formatting errors]. In this case, the calibration requirement is simply that there is no $i, j$ for which $P(R_{i, j} = 1)$ is small (i.e. the model is confident that there will be no interaction), when in fact $i$ and $j$ do interact. As mentioned above -  this is not part of this work's contributions so we are focusing on intuition rather than formalism (this argument is adapted from what we've done in followup work), but we hope this discussion shows that it is in principle possible to derive a stopping criterion in this framework.

---

### Author Response · Authors · 2024-11-26
**To all reviewers**

Dear Reviewers,

We thank you again for the thoughtful feedback on our work. We are working on incorporating changes based on the discussion in an updated version of the paper and hope to post it as soon as possible prior to the deadline.

---

> ### Author Response · Authors · 2024-11-26
> **Manuscript updated**
>
> Dear Reviewers,
>
> We thank you again for the engaging discussions and thoughtful feedback on our work. We have now updated our submission to reflect the dicussions.

---

### Meta-Review · Area_Chair_WWZR · 2024-12-18

**Metareview:**

An interesting paper, which unfortunately does not pass the bar for acceptance at ICLR. Substantial points of disagreements still coexisted with 1aqQ at the end of the review process, and while I can buy the authors' argument for point (1), I believe a bit more could have been done for points (2-3). I can only recommend that the authors polish further their paper for a next round of submission.

**Additional Comments On Reviewer Discussion:**

Substantial discussions with reviewer 1aqQ, in particular on the novelty of the framework and assumptions made, during rebuttal time.

---

### Decision · Program_Chairs · 2025-01-22

Reject